# FITS: Modeling Time Series with $10k$ Parameters

## Abstract

In this paper, we introduce FITS, a lightweight yet powerful model for time series analysis. Unlike existing models that directly process raw time-domain data, FITS operates on the principle that time series can be manipulated through interpolation in the complex frequency domain. By discarding high-frequency components with negligible impact on time series data, FITS achieves performance comparable to state-of-the-art models for time series forecasting and anomaly detection tasks, while having a remarkably compact size of only approximately $10k$ parameters. Such a lightweight model can be easily trained and deployed in edge devices, creating opportunities for various applications. The anonymous code repo is available in: `https://anonymous.4open.science/r/FITS`

## 1 Introduction

Time series analysis plays a crucial role in numerous domains, including finance, energy, weather forecasting, and signal processing, where understanding and predicting temporal patterns are essential. Existing time series analysis methods primarily focus on extracting features in the time domain (Zhou et al., 2021; Liu et al., 2022; Zeng et al., 2022; Nie et al., 2023; Zhang et al., 2022). However, due to the inherent complexity and dynamic nature of time series data, the information contained in the time domain tends to be sparse and dispersed. Consequently, researchers design intricate methodologies and complex models to capture and exploit this information, often relying on approaches such as transformer architectures (Zhou et al., 2021; Wu et al., 2021; Zhou et al., 2022a). However, these sophisticated techniques often lead to the proliferation of large-scale and computationally demanding models, posing challenges in terms of efficiency and scalability.

Conversely, the frequency domain representation of time series data offers a more concise and compact representation of its underlying information. Recognizing this potential, previous studies have explored the utilization of frequency domain information in time series analysis. For instance, FEDformer (Zhou et al., 2022a) incorporates spectral information as a supplementary feature, enhancing the modeling capabilities of transformer-based time series models. Another approach, FNet (Lee-Thorp et al., 2022), leverages frequency domain multiplication to replace convolution operations, thereby reducing computational overhead. Moreover, LTSF-Linear (Zeng et al., 2022) has demonstrated that highly accurate predictions can be achieved by solely learning the dominant periodicity. Similarly, methods like TimesNet (Wu et al., 2023) segment the time series based on frequencies with high amplitude and employ CNNs for multi-periodicity feature extraction.

However, existing methodologies often overlook the fundamental nature of the frequency domain representation, which utilizes complex numbers to express both amplitude and phase information. Motivated by the fact that longer time series segments provide a higher-resolution frequency representation, we propose FITS (**F**requency **I**nterpolation **T**ime **S**eries Analysis Baseline). The core component of FITS is a complex-valued linear layer that can explicitly learn amplitude scaling and phase shift to perform interpolation in the complex frequency domain. Although FITS conducts interpolation in the frequency domain, it remains an end-to-end time domain model incorporating

the rFFT (Brigham & Morrow, 1967). Specifically, we project the input segment to the complex frequency domain for frequency interpolation using rFFT. We then project the interpolated frequency representation back to the time domain as a longer segment for supervision. This end-to-end design enables FITS to adapt to various downstream tasks with commonly-used time domain supervision, such as forecasting and reconstruction.

Additionally, FITS incorporates a low-pass filter to obtain a compact representation with minimal information loss, resulting in small model volume and minimal computational overhead while maintaining state-of-the-art (SOTA) performance. Notably, under most settings, FITS achieves SOTA performance with under **10k parameters**, which is **50 times smaller** than the lightweight temporal linear model DLinear (Zeng et al., 2022) and approximately **10,000 times smaller** than other mainstream models. The low memory and computation overhead make FITS suitable for deploying or even training on edge devices for forecasting or anomaly detection.

To summarize, our contributions are twofold:

- We introduce FITS, a lightweight model containing merely **5k∼10k** parameters for time series analysis. Despite its compact size which is several orders of magnitude smaller than mainstream models, FITS delivers exceptional performance in various tasks, including long-term forecasting and anomaly detection, achieving state-of-the-art performance in several datasets.
- FITS employs the complex-valued neural network for time series analysis, which provides a novel perspective that simultaneously captures amplitude and phase information, leading to more comprehensive and efficient modeling of time series data.

## 2 Related Work and Motivation

### 2.1 Frequency-aware Time Series Analysis Models

Recent advancements in time series analysis have witnessed the utilization of frequency domain information to capture and interpret underlying patterns. FNet (Lee-Thorp et al., 2022) leverages a pure attention-based architecture to efficiently capture temporal dependencies and patterns solely in the frequency domain, eliminating the need for convolutional or recurrent layers. On the other hand, FEDFormer (Zhou et al., 2022a) and FiLM (Zhou et al., 2022b) incorporate frequency information as supplementary features to enhance the model's capability in capturing long-term periodic patterns and speed up computation.

The other line of work aims to capture the periodicity inherent in the data. For instance, DLinear (Zeng et al., 2022) adopts a single linear layer to extract the dominant periodicity from the temporal domain and surpasses a range of deep feature extraction-based methods. More recently, TimesNet (Wu et al., 2023) achieves state-of-the-art results by identifying several dominant frequencies instead of relying on a single dominant periodicity. Specifically, they use the Fast Fourier Transform (FFT) to find the frequencies with the largest energy and reshape the original 1D time series into 2D images according to their periods.

However, these approaches still rely on feature engineering to identify the dominant period set. Selecting this set based on energy may only consider the dominant period and its harmonics, limiting the information captured. Moreover, these methodologies are still considered inefficient and prone to overfitting.

### 2.2 Divide and Conquer the Frequency Components

Treating a time series as a signal allows us to break it down into a linear combination of sinusoidal components without any information loss. Each component possesses a unique frequency, initial phase, and amplitude. Forecasting directly on the original time series can be challenging, but forecasting each frequency component is comparatively straightforward, as we only need to apply a phase bias to the sinusoidal wave based on the time shift. Subsequently, we linearly combine these shifted sinusoidal waves to obtain the forecasting result.

This approach effectively preserves the frequency characteristics of the given look-back window while maintaining semantic consistency between the look-back window and the forecasting horizon.

89 Specifically, the resulting forecasted values maintain the frequency features of the original time series
90 with a reasonable time shift, ensuring that semantic consistency is maintained.

91 However, forecasting each sinusoidal component in the time domain can be cumbersome, as the
92 sinusoidal components are treated as a sequence of data points. To address this, we propose conducting
93 this manipulation in the complex frequency domain, which offers a more compact and information-
94 rich representation, as described below.

# 3 Method

## 3.1 Preliminary: FFT and Complex Frequency Domain

97 The Fast Fourier Transform (FFT, (Brigham & Morrow, 1967)) is a widely used algorithm for
98 efficiently computing the Discrete Fourier Transform (DFT) of a sequence of complex numbers. The
99 DFT is a mathematical operation that converts a discrete-time signal from the time domain to the
100 complex frequency domain. In cases where the input signal is real, such as in time series analysis,
101 the Real FFT (rFFT) is commonly used to obtain a compact representation. With an input of $N$ real
102 numbers, the rFFT produces a sequence of $N/2 + 1$ complex numbers that represent the signal in the
103 complex frequency domain.

**Complex Frequency Domain**

105 In Fourier analysis, the complex frequency domain is a representation of a signal in which each
106 frequency component is characterized by a complex number. This complex number captures both
107 the amplitude and phase of the component, providing a comprehensive description. The amplitude
108 of a frequency component represents the magnitude or strength of that component in the original
109 time-domain signal. In contrast, the phase represents the temporal shift or delay introduced by that
110 component. Mathematically, the complex number associated with a frequency component can be
111 represented as a complex exponential element with a given amplitude and phase:

$$X(f) = |X(f)|e^{j\theta(f)},$$

112 where $X(f)$ is the complex number associated with the frequency component at frequency $f$, $|X(f)|$
113 is the amplitude of the component, and $\theta(f)$ is the phase of the component. As shown in Fig. 1(a), in
114 the complex plane, the complex exponential element can be visualized as a vector with a length equal
115 to the amplitude and angle equal to the phase:

$$X(f) = |X(f)|(\cos\theta(f) + j\sin\theta(f))$$

116 Therefore, the complex number in the complex frequency domain provides a concise and elegant
117 means of representing the amplitude and phase of each frequency component in the Fourier transform.

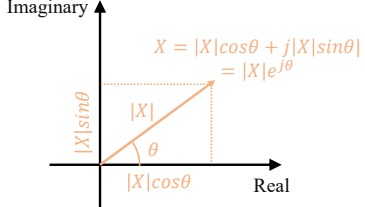

(a) Complex number on the complex plane

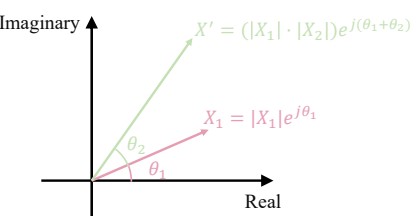

(b) Complex number multiplication

Figure 1: Illustration of Complex Number Visualization and Multiplication

118 **Time Shift and Phase Shift**. The time shift of a signal corresponds to the phase shift in the frequency
119 domain. Especially in the complex frequency domain, we can express such phase shift by multiplying
120 a unit complex exponential element with the corresponding phase. Mathematically, if we shift a
121 signal $x(t)$ forward in time by a constant amount $\tau$, resulting in the signal $x(t - \tau)$, the Fourier
122 transform is given by:

$$X_\tau(f) = e^{-j2\pi f\tau}X(f) = |X(f)|e^{j(\theta(f)-2\pi f\tau)} = [cos(-2\pi f\tau) + jsin(-2\pi f\tau)]X(f)$$

The shifted signal still has an amplitude of $|X(f)|$, while the phase $\theta_\tau(f) = \theta(f) - 2\pi f\tau$ shows a shift which is linear to the time shift.

In summary, the amplitude scaling and phase shifting can be simultaneously expressed as the multiplication of complex numbers, as shown in Fig. 1(b).

## 3.2 FITS Pipeline

Motivated by the fact that a longer time series provides a higher frequency resolution in its frequency representation, we train FITS to generate an extended time series segment by interpolating the frequency representation of the input time series segment. We use a complex-valued linear layer to learn such interpolation. According to the fact that the amplitude scaling and phase shifting can be conveniently expressed as the multiplication of complex numbers, such complex linear combination allows FITS to effectively incorporate both the amplitude scaling and phase shift of frequency components during the interpolation process. As shown in Fig. 2, we use rFFT to project time series segments to the complex frequency domain. After the interpolation, the frequency representation is projected back with inverse rFFT (irFFT).

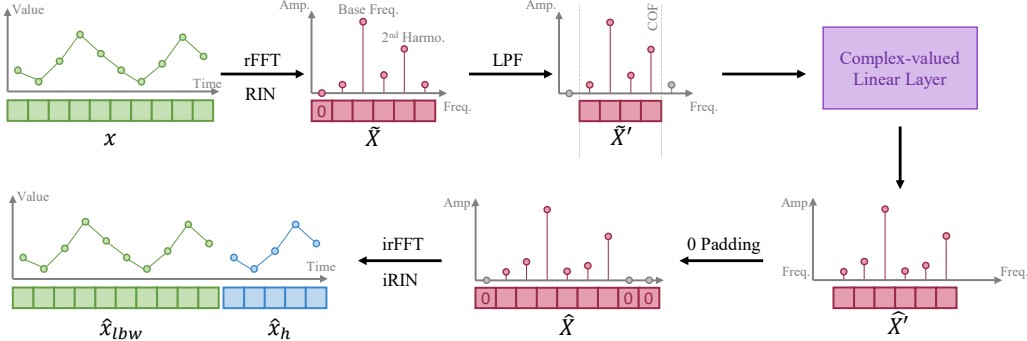

Figure 2: Pipeline of FITS, with a focus on the forecasting task. The reconstruction task follows the same pipeline, except for the reconstruction supervision loss.

However, we cannot directly use the frequency representation of the original input time series segment because the mean of such segments will result in a very large 0-frequency component in its complex frequency representation. To eliminate the 0-frequency component, we pass it through reversible instance-wise normalization (RIN) (Kim et al., 2022) to obtain a zero-mean instance. As a result, the normalized complex frequency representation now has a length of $N/2$, where $N$ represents the original length of the time series.

Furthermore, we incorporate a low-pass filter (LPF) into the FITS model to further reduce its size. The LPF removes high-frequency components above a specified cutoff frequency, resulting in a more compact model representation while retaining the important information of the time series. The rationale behind this design will be elaborated in the subsequent section. Despite operating in the frequency domain, FITS is supervised in the time domain using common loss functions such as Mean Squared Error (MSE) after the irFFT, allowing for diverse supervision tailored to different time series downstream tasks.

In the case of forecasting tasks, we generate the look-back window along with the horizon as shown in Fig. 2. This allows us to provide supervision for forecasting and backcasting, where the model is encouraged to accurately reconstruct the look-back window. Our ablation study reveals that combining backcast and forecast supervision can yield improved performance in certain scenarios.

For reconstruction tasks, we downsample the original time series segment based on a specific downsampling rate. Subsequently, FITS is employed to perform frequency interpolation, enabling the reconstruction of the downsampled segment back to its original form. Thus, direct supervision is applied using reconstruction loss to ensure faithful reconstruction. The reconstruction tasks also follow the pipeline in Fig. 2 with the supervision replaced with reconstruction loss.

### 3.3 Key Mechanisms of FITS

**Complex Frequency Linear Interpolation.** To control the output length of the model, we introduce an interpolation rate denoted as $\eta$, which represents the ratio of the model's output length $L_o$ to its corresponding input length $L_i$.

It is worth noting that frequency interpolation operates on the normalized complex frequency representation, which has half the length of the original time series. Importantly, this interpolation rate can also be applied to the frequency domain, as indicated by the equation:

$$\eta_{freq} = \frac{L_o/2}{L_i/2} = \frac{L_o}{L_i} = \eta$$

Based on this formula, with an arbitrary frequency $f$, the frequency band $1 \sim f$ in the original signal is linearly projected to the frequency band $1 \sim \eta f$ in the output signal. As a result, we define the input length of our complex-valued linear layer as $L$ and the interpolated output length as $\eta L$. Notably, when applying the Low Pass Filter (LPF), the value of $L$ corresponds to the cutoff frequency (COF) of the LPF. After performing frequency interpolation, the complex frequency representation is zero-padded to a length of $L_o/2$, where $L_o$ represents the desired output length. Prior to applying the irFFT, an additional zero is introduced as the representation's zero-frequency component.

**Low Pass Filter (LPF).** The primary objective of incorporating the LPF within FITS is to compress the model's volume while preserving essential information. The LPF achieves this by discarding frequency components above a specified cutoff frequency (COF), resulting in a more concise frequency domain representation. The LPF retains the relevant information in the time series while discarding components beyond the model's learning capability. This ensures that a significant portion of the original time series' meaningful content is preserved. As demonstrated in Fig. 3, the filtered waveform exhibits minimal distortion even when only preserving a quarter of the original frequency domain representation. Furthermore, the high-frequency components filtered out by the LPF typically comprise noise and trends, which are inherently irrelevant for effective time series modeling.

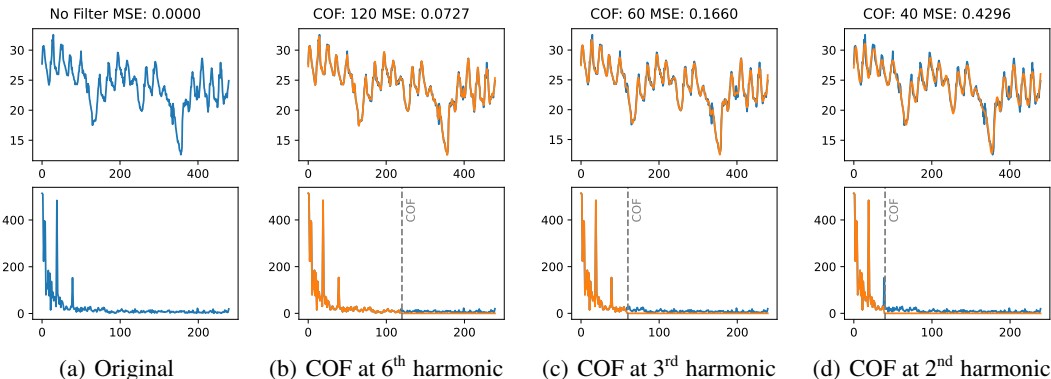

Figure 3: Waveform (1st row) and amplitude spectrum (2nd row) of a time series segment selected from the 'OT' channel of the ETTh1 dataset, spanning from the 1500th to the 1980th data point. The segment has a length of 480, and its dominant periodicity is 24, corresponding to a base frequency of 20. The blue lines represent the waveform/spectrum with no applied filter, while the orange lines represent the waveform/spectrum with the filter applied. The filter cutoff frequency is chosen based on a harmonic of the original time series.

Selecting an appropriate cutoff frequency (COF) remains a nontrivial challenge. To address this, we propose a method based on the harmonic content of the dominant frequency. Harmonics, which are integer multiples of the dominant frequency, play a significant role in shaping the waveform of a time series. By aligning the cutoff frequency with these harmonics, we keep relevant frequency components associated with the signal's structure and periodicity. This approach leverages the inherent relationship between frequencies to extract meaningful information while suppressing noise and irrelevant high-frequency components. The impact of COF on different harmonics' waveforms is shown in Fig. 3. We further elaborate on the impact of COF in our experimental results.

## 4  Experiments for Forecasting

### 4.1  Forecasting as Frequency Interpolation

Typically, the forecasting horizon is shorter than the given look-back window, rendering direct interpolation unsuitable. Instead, we formulate the forecasting task as the interpolation of a look-back window, with length $L$, to a combination of the look-back window and forecasting horizon, with length $L + H$. This design enables us to provide more supervision during training. With this approach, we can supervise not only the forecasting horizon but also the backcast task on the look-back window. Our experimental results demonstrate that this unique training strategy contributes to the improved performance of FITS. The interpolation rate of the forecasting task is calculated by:

$$\eta_{Fore} = 1 + \frac{H}{L},$$

where $L$ represents the length of the look-back window and $H$ represents the length of the forecasting horizon.

### 4.2  Experiment Settings

**Datasets.** All datasets used in our experiments are widely-used and publicly available real-world datasets, including, Traffic, Electricity, Weather, ETT (Zhou et al., 2021). We summarize the characteristics of these datasets in Tab. 1. Apart from these datasets for long-term time series forecasting, we also use the M4 dataset to test the short-term forecasting performance.

Table 1: The statistics of the seven used forecasting datasets.

| Dataset | Traffic | Electricity | Weather | ETTh1&ETTh2 | ETTm1 &ETTm2 |
|---|---|---|---|---|---|
| Channels | 862 | 321 | 21 | 7 | 7 |
| Sampling Rate | 1hour | 1hour | 10min | 1hour | 15min |
| Total Timesteps | 17,544 | 26,304 | 52,696 | 17,420 | 69,680 |

**Baselines**. To evaluate the performance of FITS in comparison to state-of-the-art time series forecasting models, including PatchTST (Nie et al., 2023), TimesNet (Wu et al., 2023), FEDFormer (Zhou et al., 2022a), FiLM (Zhou et al., 2022b) and LTSF-Linear (Zeng et al., 2023), we directly refer to the reported results in the original papers under the same settings. We report the comparison with other transformer-based methods in the appendix.

**Evaluation metrics**. We follow the previous works (Zhou et al., 2022a; Zeng et al., 2022; Zhang et al., 2022) to compare forecasting performance using Mean Squared Error (MSE) as the core metrics. Moreover, to evaluate the short-term forecasting, we symmetric Mean Absolute Percentage Error (SMAPE) following TimesNet (Wu et al., 2023).

**Implementation details**. Following the settings of LTSF-Linear (Zeng et al., 2023), we set the look-back window of FITS as 720 for any forecasting horizon. Further experiments also show that a longer look-back window can result in better performance. To avoid information leakage, We choose the hyper-parameter based on the performance of the validation set.

### 4.3  Comparisons with SOTAs

**Competitive Performance with High Efficiency**

We present the results of our experiments on long-term forecasting in Tab. 2 and Tab. 3. The results for short-term forecasting on the M4 dataset are provided in the Appendix. Remarkably, our FITS consistently achieves comparable or even superior performance across all experiments.

Tab. 4 presents the number of trainable parameters for various TSF models using a look-back window of 96 and a forecasting horizon of 720 on the Electricity dataset. The table clearly demonstrates the exceptional efficiency of FITS compared to other models.

Among the listed models, the parameter counts range from millions down to thousands. Notably, large models such as TimesNet and Pyraformer require a staggering number of parameters, with

Table 2: Long-term forecasting results on ETT dataset in MSE. The best result is highlighted in **bold**, and the second best is highlighted with underline. IMP is the improvement between FITS and the second best/ best result, where a larger value indicates a better improvement.

| Dataset | ETTh1 | | | | ETTh2 | | | | ETTm1 | | | | ETTm2 | | | |
|---|---|---|---|---|---|---|---|---|---|---|---|---|---|---|---|---|
| Horizon | 96 | 192 | 336 | 720 | 96 | 192 | 336 | 720 | 96 | 192 | 336 | 720 | 96 | 192 | 336 | 720 |
| PatchTST | **0.370** | 0.413 | **0.422** | 0.447 | **0.274** | 0.341 | **0.329** | 0.379 | **0.293** | **0.333** | 0.369 | **0.416** | 0.166 | 0.223 | 0.274 | 0.362 |
| TimesNet | 0.384 | 0.436 | 0.491 | 0.521 | 0.340 | 0.402 | 0.452 | 0.462 | 0.338 | 0.374 | 0.410 | 0.478 | 0.187 | 0.249 | 0.321 | 0.408 |
| FEDFormer | 0.376 | 0.420 | 0.459 | 0.506 | 0.346 | 0.429 | 0.496 | 0.463 | 0.379 | 0.426 | 0.445 | 0.543 | 0.203 | 0.269 | 0.325 | 0.421 |
| FiLM | 0.371 | 0.414 | 0.442 | 0.465 | 0.284 | 0.357 | 0.377 | 0.439 | 0.302 | 0.338 | 0.373 | 0.420 | 0.165 | 0.222 | 0.277 | 0.371 |
| Dlinear | 0.374 | **0.405** | 0.429 | 0.440 | 0.338 | 0.381 | 0.400 | 0.436 | 0.299 | 0.335 | 0.369 | 0.425 | 0.167 | 0.221 | 0.274 | 0.368 |
| FITS | 0.375 | 0.408 | 0.429 | **0.427** | 0.274 | 0.333 | 0.340 | 0.374 | 0.305 | 0.339 | **0.367** | 0.418 | **0.164** | **0.217** | **0.269** | **0.347** |
| IMP | -0.005 | -0.003 | -0.007 | 0.013 | 0 | 0.008 | -0.011 | 0.005 | -0.012 | -0.006 | 0.002 | -0.002 | 0.002 | 0.004 | 0.005 | 0.015 |

Table 3: Long-term forecasting results on three popular datasets in MSE. The best result is highlighted in **bold** and the second best is highlighted with underline. IMP is the improvement between FITS and the second best/ best result, where a larger value indicates a better improvement.

| Dataset | Electricity | | | | Traffic | | | | Weather | | | |
|---|---|---|---|---|---|---|---|---|---|---|---|---|
| Horizon | 96 | 192 | 336 | 720 | 96 | 192 | 336 | 720 | 96 | 192 | 336 | 720 |
| PatchTST | **0.129** | **0.147** | **0.163** | **0.197** | **0.360** | **0.379** | **0.392** | **0.432** | 0.149 | 0.194 | 0.245 | 0.314 |
| TimesNet | 0.168 | 0.184 | 0.198 | 0.220 | 0.593 | 0.617 | 0.629 | 0.640 | 0.172 | 0.219 | 0.280 | 0.365 |
| FEDFormer | 0.193 | 0.201 | 0.214 | 0.246 | 0.587 | 0.604 | 0.621 | 0.626 | 0.217 | 0.276 | 0.339 | 0.403 |
| FiLM | 0.154 | 0.164 | 0.188 | 0.236 | 0.416 | 0.408 | 0.425 | 0.520 | 0.199 | 0.228 | 0.267 | 0.319 |
| Dlinear | 0.140 | 0.153 | 0.169 | 0.203 | 0.410 | 0.423 | 0.435 | 0.464 | 0.176 | 0.218 | 0.262 | 0.323 |
| FITS | 0.138 | 0.152 | 0.166 | 0.205 | 0.401 | 0.407 | 0.420 | 0.456 | **0.145** | **0.188** | **0.236** | **0.308** |
| IMP | -0.009 | -0.005 | -0.003 | -0.008 | -0.041 | -0.028 | -0.028 | -0.024 | 0.004 | 0.006 | 0.009 | 0.006 |

300.6M and 241.4M, respectively. Similarly, popular models like Transformer, Informer, Autoformer, and FEDformer have parameter counts in the range of 13.61M to 20.68M. Even the lightweight yet state-of-the-art model PatchTST has a parameter count of over 1 million.

In contrast, FITS stands out as a highly efficient model with an impressively low parameter count. With only 4.5K to 16K parameters, FITS achieves comparable or even superior performance compared to these larger models. It is worth highlighting that FITS requires significantly fewer parameters compared to the next smallest model, Dlinear, which has 139.7K parameters. For instance, when considering a 720 look-back window and a 720 forecasting horizon, the Dlinear model requires over 1 million parameters, whereas FITS achieves similar performance with only 10k-50k parameters.

This analysis showcases the remarkable efficiency of FITS. Despite its small size, FITS consistently achieves competitive results, making it an attractive option for time series analysis tasks. FITS demonstrates that achieving state-of-the-art or close to state-of-the-art performance with a considerably reduced parameter footprint is possible, making it an ideal choice for resource-constrained environments.

Table 4: Number of trainable parameters and MACs of TSF models under look-back window=96 and forecasting horizon=720 on the Electricity dataset.

| Model | Parameters | MACs |
|---|---|---|
| TimesNet | 301.7M | 1226.49G |
| Pyraformer | 241.4M | 0.80G |
| Transformer | 13.61M | 4.03G |
| Informer | 14.38M | 3.93G |
| Autoformer | 14.91M | 4.41G |
| FiLM | 14.91M | 5.97G |
| FEDformer | 20.68M | 4.41G |
| PatchTST | 1.5M | 5.07G |
| DLinear | 139.7K | 40M |
| FITS (Ours) | **4.5K~10K** | **1.6M~8.9M** |

**Case Study on ETTh2 Dataset**

We conduct a comprehensive case study on the performance of FITS using the ETTh2 dataset, which further highlights the impact of the look-back window and cutoff frequency on model performance. We provide a case study on other datasets in the Appendix. In our experiments, we observe that increasing the look-back window generally leads to improved performance, while the effect of increasing the cutoff frequency is minor.

Tab. 5 showcases the performance results obtained with different look-back window sizes and cutoff frequencies. Larger look-back windows tend to yield better performance across the board. On the other hand, increasing the cutoff frequency only results in marginal performance improvements. However, it is important to note that higher cutoff frequencies come at the expense of increased computational resources, as illustrated in Tab. 6.

Table 5: The results on the ETTh2 dataset. Values are visualized with a green background, where darker background indicates worse performance. The top-5 best results are highlighted with a red background, and the absolute best result is highlighted with **red bold** font. **F** represents supervision on the forecasting task, while **B+F** represents supervision on backcasting and forecasting tasks.

| | Look-back Window | 90 | | 180 | | 360 | | 720 | |
|---|---|---|---|---|---|---|---|---|---|
| Horizon | COF/nth Harmonic | F | B+F | F | B+F | F | B+F | F | B+F |
| 96 | 2 | 0.297687 | 0.296042 | 0.291606 | 0.289387 | 0.278644 | 0.278403 | 0.277708 | 0.27696 |
| | 3 | 0.297796 | 0.297377 | 0.290061 | 0.288239 | 0.277512 | 0.277746 | 0.276537 | 0.277068 |
| | 4 | 0.297106 | 0.295624 | 0.290725 | 0.287993 | 0.27624 | 0.27693 | **0.274207** | 0.274498 |
| | 5 | 0.296168 | 0.296698 | 0.288518 | 0.287375 | 0.276367 | 0.277935 | 0.275989 | 0.275636 |
| 192 | 2 | 0.380163 | 0.379868 | 0.360591 | 0.359769 | 0.336552 | 0.337976 | 0.334854 | 0.335887 |
| | 3 | 0.37983 | 0.381802 | 0.359088 | 0.359498 | 0.336384 | 0.336358 | 0.334666 | 0.335507 |
| | 4 | 0.379657 | 0.380439 | 0.359087 | 0.358536 | 0.334803 | 0.349995 | 0.333522 | **0.333382** |
| | 5 | 0.378556 | 0.379883 | 0.358809 | 0.359376 | 0.335451 | 0.343227 | 0.33384 | 0.335053 |
| 336 | 2 | 0.402706 | 0.404805 | 0.373257 | 0.374678 | 0.344241 | 0.344414 | 0.341869 | 0.342549 |
| | 3 | 0.403238 | 0.404878 | 0.372231 | 0.373948 | 0.345578 | 0.344976 | 0.341436 | 0.342793 |
| | 4 | 0.402702 | 0.407712 | 0.376199 | 0.374435 | 0.343004 | 0.344167 | **0.340795** | 0.342245 |
| | 5 | 0.403484 | 0.409516 | 0.375102 | 0.37462 | 0.344333 | 0.342731 | 0.341043 | 0.342214 |
| 720 | 2 | 0.420072 | 0.424272 | 0.403985 | 0.407392 | 0.379822 | 0.38519 | 0.376871 | 0.37677 |
| | 3 | 0.418323 | 0.420538 | 0.400986 | 0.40686 | 0.379638 | 0.386397 | 0.376236 | 0.376004 |
| | 4 | 0.417485 | 0.420982 | 0.399987 | 0.408128 | 0.379096 | 0.386409 | 0.375865 | 0.375637 |
| | 5 | 0.419122 | 0.420355 | 0.400776 | 0.407871 | 0.378665 | 0.390754 | 0.377138 | **0.374586** |

Considering these observations, we find utilizing a longer look-back window in combination with a low cutoff frequency to achieve near state-of-the-art performance with minimal computational cost. For instance, FITS surpasses other methods when employing a 720 look-back window and setting the cutoff frequency to the second harmonic. Remarkably, FITS achieves state-of-the-art performance with a parameter count of only around 10k. Moreover, by reducing the look-back window to 360, FITS already achieves close-to-state-of-the-art performance by setting the cutoff frequency to the second harmonic, resulting in a further reduction of the model's parameter count to under 5k (as shown in Tab. 6).

Table 6: The number of parameters under different settings on ETTh1 & ETTh2 dataset.

| | | Look-back Window | | | |
|---|---|---|---|---|---|
| Horizon | COF/nth Harmonic | 90 | 180 | 360 | 720 |
| 96 | 2 | 703 | 1053 | 2279 | 5913 |
| | 3 | 1035 | 1820 | 4307 | 12064 |
| | 4 | 1431 | 2752 | 6975 | 20385 |
| | 5 | 1922 | 3876 | 10374 | 31042 |
| 192 | 2 | 1064 | 1431 | 2752 | 6643 |
| | 3 | 1564 | 2450 | 5192 | 13520 |
| | 4 | 2187 | 3698 | 8475 | 22815 |
| | 5 | 2914 | 5253 | 12558 | 34694 |
| 336 | 2 | 1615 | 1998 | 3483 | 7665 |
| | 3 | 2392 | 3395 | 6608 | 15704 |
| | 4 | 3321 | 5160 | 10725 | 26460 |
| | 5 | 4402 | 7293 | 15834 | 40006 |
| 720 | 2 | 3078 | 3510 | 5418 | 10512 |
| | 3 | 4554 | 5950 | 10266 | 21424 |
| | 4 | 6318 | 9030 | 16650 | 36180 |
| | 5 | 8370 | 12750 | 24570 | 54780 |

These results emphasize the lightweight nature of FITS, making it highly suitable for deployment and training on edge devices with limited computational resources. By carefully selecting the look-back window and cutoff frequency, FITS can achieve excellent performance while maintaining computational efficiency, making it an appealing choice for real-world applications.

## 5 Experiment for Anomaly Detection

### 5.1 Reconstruction as Frequency Interpolation

As discussed before, we tackle the anomaly detection tasks in the self-supervised reconstructing approach. Specifically, we make a $N$ time down-sampling on the input and train a FITS network with an interpolation rate of $\eta_{Rec} = N$ to up-sample it.

### 5.2 Experiment Settings

**Datasets**. We use five commonly used benchmark datasets: SMD (Server Machine Dataset (Su et al., 2019)), PSM (Polled Server Metrics (Abdulaal et al., 2021)), SWaT (Secure Water Treatment (Mathur & Tippenhauer, 2016)), MSL (Mars Science Laboratory rover), and SMAP (Soil Moisture Active Passive satellite) (Hundman et al., 2018).

**Baselines**. We compare FITS with models such as TimesNet (Wu et al., 2023), Anomaly Transformer (Xu et al., 2022), THOC (Shen et al., 2020), Omnianomaly (Su et al., 2019). Following TimesNet (Wu et al., 2023), we also compare the anomaly detection performance with other models (Zeng et al., 2023; Zhang et al., 2022; Woo et al., 2022; Zhou et al., 2022a).

**Evaluation metrics**. Following the previous works (Xu et al., 2022; Shen et al., 2020; Wu et al., 2023), we use Precision, Recall, and F1-score as metrics.

**Implementation details**. We use a window size of 200 and downsample the time series segment by a factor of 4 to match the original segment during training with the FITS model. Anomaly detection follows the methodology of the Anomaly Transformer (Xu et al., 2022), where time points exceeding a certain reconstruction loss threshold are classified as anomalies. The threshold is selected based on the highest F1 score achieved on the validation set. To handle consecutive abnormal segments, we adopt a widely-used adjustment strategy (Su et al., 2019; Xu et al., 2018; Shen et al., 2020), considering all anomalies within a specific successive abnormal segment as correctly detected when one anomalous time point is identified. This approach aligns with real-world applications, where an abnormal time point often triggers the attention to the entire segment.

Table 7: Anomaly detection result of F1-scores on 5 datasets. The best result is highlighted in **bold**, and the second best is highlighted with underline. Full results are reported in the Appendix.

| Models | FITS | TimesNet | Anomaly Transformer | THOC | Omni Anomaly | Stationary Transformer | LightTS | Dlinear | IMP |
|--------|------|----------|---------------------|------|--------------|------------------------|---------|---------|-----|
| SMD | **99.95** | 85.81 | 92.33 | 84.99 | 85.22 | 84.72 | 82.53 | 77.1 | 7.62 |
| PSM | 93.96 | 97.47 | 97.89 | **98.54** | 80.83 | 97.29 | 97.15 | 93.55 | -3.93 |
| SWaT | **98.9** | 91.74 | 94.07 | 85.13 | 82.83 | 79.88 | 93.33 | 87.52 | 4.83 |
| SMAP | 70.74 | 71.52 | **96.69** | 90.68 | 86.92 | 71.09 | 69.21 | 69.26 | -25.95 |
| MSL | 78.12 | 85.15 | **93.59** | 89.69 | 87.67 | 77.5 | 78.95 | 84.88 | -15.47 |

## 5.3 Comparisons with SOTAs

As shown in Tab. 7, FITS achieves remarkable results on several datasets. Notably, on the SMD and SWaT datasets, FITS exhibits exceptional performance with F1-scores almost reaching perfection at around 99.95% and 98.9%, respectively. This demonstrates FITS' ability to accurately detect anomalies and classify them correctly. In comparison, other models, such as TimesNet, Anomaly Transformer, and Stationary Transformer, struggle to match FITS' performance on these datasets.

However, FITS shows comparatively lower performance on the SMAP and MSL datasets. These datasets present a challenge due to their binary event data nature, which may not be effectively captured by FITS' frequency domain representation. While models specifically designed for anomaly detection, such as THOC and Omni Anomaly, achieve higher F1-scores on these datasets.

For a more comprehensive evaluation, waveform visualizations and detailed analysis can be found in the appendix, providing deeper insights into FITS' strengths and limitations in different anomaly detection scenarios. It is important to note that the reported results are achieved with a parameter range of 1-4K and MACs (Multiply-Accumulate Operations) of 10-137K, which will be further detailed in the appendix.

## 6 Conclusions and Discussion

In this paper, we propose FITS for time series analysis, a low-cost model with $10k$ parameters that can achieve performance comparable to state-of-the-art models that are often several orders of magnitude larger. As a frequency-domain modeling technique, FITS has difficulty handling binary-valued time series and time series with missing data. For the former category, time-domain modeling is preferable as the raw data format is sufficiently compact. For the latter category, we could first employ simple yet effective time-domain imputation techniques and then apply FITS for efficient analysis.

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
