# FITS: Modeling Time Series with $10k$ Parameters Supplementary Material

## A    Pipeline for Reconstruction

The pipeline for the reconstruction task is shown in Fig. 1. Note that the input is a downsampled time series segment, and the output is supervised on the original time series segment.

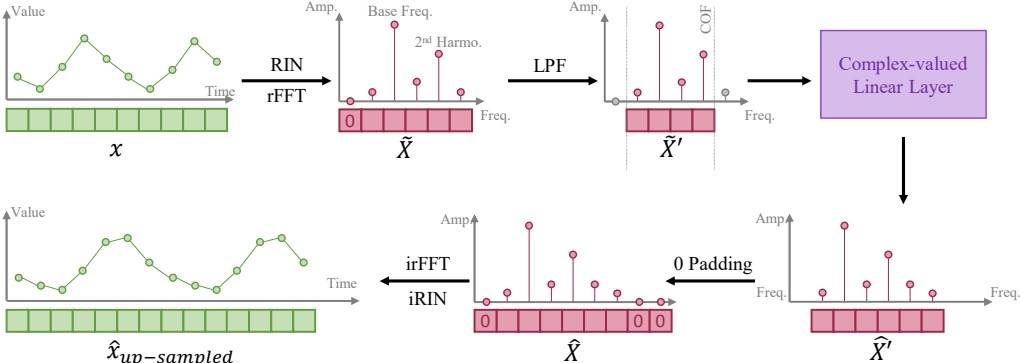

Figure 1: Pipeline of FITS, with a focus on the Reconstruction task.

## B    More Results on Forecasting Task

We show the comparison with transformer-based models, short-term forecasting on M4, and the impact of random seeds below.

### B.1    Comparison with Transformer-based Methods

We further compare FITS with Autoformer (Wu et al., 2021), Informer (Zhou et al., 2021) and Pyraformer (Liu et al., 2022). The results are shown in Tab. 1 and Tab. 2.

Table 1: Long-term forecasting results on ETT datasets in MSE. The best result is highlighted in **bold**.

| Dataset | ETTh1 | | | | ETTh2 | | | | ETTm1 | | | | ETTm2 | | | |
|---|---|---|---|---|---|---|---|---|---|---|---|---|---|---|---|---|
| Horizon | 96 | 192 | 336 | 720 | 96 | 192 | 336 | 720 | 96 | 192 | 336 | 720 | 96 | 192 | 336 | 720 |
| Autoformer | 0.449 | 0.500 | 0.521 | 0.514 | 0.358 | 0.456 | 0.482 | 0.515 | 0.505 | 0.553 | 0.621 | 0.671 | 0.255 | 0.281 | 0.339 | 0.433 |
| Informer | 0.865 | 1.008 | 1.107 | 1.181 | 3.755 | 5.602 | 4.721 | 3.647 | 0.672 | 0.795 | 1.212 | 1.166 | 0.365 | 0.533 | 1.363 | 3.379 |
| FEDFormer | 0.376 | 0.420 | 0.459 | 0.506 | 0.346 | 0.429 | 0.496 | 0.463 | 0.379 | 0.426 | 0.445 | 0.543 | 0.203 | 0.269 | 0.325 | 0.421 |
| Pyraformer | 0.664 | 0.790 | 0.891 | 0.963 | 0.645 | 0.788 | 0.907 | 0.963 | 0.543 | 0.557 | 0.754 | 0.908 | 0.435 | 0.730 | 1.201 | 3.625 |
| FITS | **0.375** | **0.408** | **0.429** | **0.427** | **0.274** | **0.333** | **0.340** | **0.374** | **0.305** | **0.339** | **0.367** | **0.418** | **0.164** | **0.217** | **0.269** | **0.347** |

Table 2: Long-term forecasting results on three popular datasets in MSE. The best result is highlighted in **bold**.

| Dataset | Electricity | | | | Traffic | | | | Weather | | | |
|---|---|---|---|---|---|---|---|---|---|---|---|---|
| Horizon | 96 | 192 | 336 | 720 | 96 | 192 | 336 | 720 | 96 | 192 | 336 | 720 |
| Autoformer | 0.201 | 0.222 | 0.231 | 0.254 | 0.613 | 0.616 | 0.622 | 0.660 | 0.266 | 0.307 | 0.359 | 0.419 |
| Informer | 0.274 | 0.296 | 0.300 | 0.373 | 0.719 | 0.696 | 0.777 | 0.864 | 0.300 | 0.598 | 0.578 | 1.059 |
| FEDFormer | 0.193 | 0.201 | 0.214 | 0.246 | 0.587 | 0.604 | 0.621 | 0.626 | 0.217 | 0.276 | 0.339 | 0.403 |
| Pyraformer | 0.386 | 0.386 | 0.378 | 0.376 | 2.085 | 0.867 | 0.869 | 0.881 | 0.896 | 0.622 | 0.739 | 1.004 |
| FITS | **0.138** | **0.152** | **0.166** | **0.205** | **0.401** | **0.407** | **0.420** | **0.456** | **0.145** | **0.188** | **0.236** | **0.308** |

Table 3: Results on M4 dataset in SMAPE.

| | FITS | DLinear | TimesNet | N-Hits | N-Beats |
|---|---|---|---|---|---|
| Yearly | 14.00 | 16.96 | 13.38 | 13.41 | 13.43 |
| Quarterly | 10.72 | 12.14 | 10.1 | 10.2 | 10.12 |
| Monthly | 13.49 | 13.51 | 12.67 | 12.7 | 12.67 |

## B.2 Short-term Forecasting on M4

We evaluate FITS' performance on the M4 dataset following the TimesNet (Wu et al., 2023). We retrieve the following results from the TimesNet paper. As shown in Tab.3, FITS shows the suboptimal results on the M4 dataset. The reason for this outcome is threefold. First, the M4 dataset is a collection of many time series from different domains. These time series have different temporal information and periodicity, and no correlations exist among them. We can not regard them as simple multivariate forecasting tasks. Second, other models have a very large amount of parameters, especially TimesNet, which makes them have enough capability to model such diverse datasets with one model. However, considering the lightweight of FITS, it is hard for it to achieve ideal results. Finally, the setting for the M4 dataset is not suitable for FITS. The look-back window is set to 12, 16, and 36 for yearly, quarterly, and monthly prediction accordingly, which is twice the length of the forecasting horizon. Such a short look-back window is very difficult to extract meaningful frequency representation, which further worsens the FITS' performance. We compare FITS with lightweight model DLinear (Zeng et al., 2022), state-of-the-art model TimesNet (Wu et al., 2023) and two hierarchical time series modeling model N-Hits (Challu et al., 2022) and N-Beats (Oreshkin et al., 2019).

## B.3 Impact of Random Seeds

We run the experiment 4 times with different chosen random seeds (i.e., 114, 514, 1919, 810) to get the standard deviation. As shown in Tab. 4, random seeds make very little impact on the FITS. Thanks to the small number of parameters, FITS is very robust to such noise.

Table 4: Table of mean and standard deviation (std) of FITS on ETTh2 dataset. The data is in mean(std) format.

| Horizon | COF/nth Harmonic | 90 | 180 | 360 | 720 |
|---|---|---|---|---|---|
| 96 | 2 | 0.295414(1.03e-07) | 0.290974(3.18e-07) | 0.281156(1.0334e-05) | 0.276398(1.35e-07) |
| | 3 | 0.294779(1.5e-07) | 0.297235(1.73e-04) | 0.284811(1.175e-04) | 0.27579(2.088e-06) |
| | 4 | 0.29443(6.7e-08) | 0.289059(8.5e-08) | 0.27689(9.2e-08) | 0.2733(1.5e-08) |
| | 5 | 0.294409(1.194e-06) | 0.291449(1.5622e-05) | 0.277162(1.87e-07) | 0.273262(2.9e-08) |
| 192 | 2 | 0.378869(2.763e-06) | 0.361255(5.69e-07) | 0.337159(9.6e-08) | 0.334243(1e-09) |
| | 3 | 0.377842(8.3e-08) | 0.360074(2.597e-06) | 0.336451(3.1e-08) | 0.333356(4e-09) |
| | 4 | 0.377441(3.04e-07) | 0.359773(4.75e-07) | 0.335045(9e-09) | 0.332138(3e-09) |
| | 5 | 0.377859(7.4e-08) | 0.358795(2.5e-07) | 0.337158(1.2363e-05) | 0.332015(1.8e-08) |
| 336 | 2 | 0.401767(8.53e-07) | 0.37311(2.39e-07) | 0.343065(2.3e-08) | 0.341445(2.849e-06) |
| | 3 | 0.402027(6.81e-07) | 0.37219(5.6e-07) | 0.342487(7e-09) | 0.340162(2e-09) |
| | 4 | 0.403194(7.561e-06) | 0.373196(8.08e-07) | 0.341473(7e-09) | 0.33921(5e-09) |
| | 5 | 0.404614(4.802e-05) | 0.372623(5.05e-07) | 0.341668(5.9e-08) | 0.339122(1.2e-08) |
| 720 | 2 | 0.416372(6.678e-06) | 0.400324(6.35e-07) | 0.380281(4.9e-08) | 0.374877(1.09e-07) |
| | 3 | 0.414283(2.18e-07) | 0.410352(4.126e-04) | 0.379871(1.39e-07) | 0.374562(6e-09) |
| | 4 | 0.415606(2.705e-06) | 0.398548(2.17e-07) | 0.378965(5.4e-08) | 0.373264(2.8e-08) |
| | 5 | 0.414254(4.21e-07) | 0.401551(2.6431e-05) | 0.378391(9e-09) | 0.373222(3e-09) |

## C   Case Study on Other Datasets

We show the parameter table and performance on other datasets below.

### C.1   ETTm1 & m2

Tab. 5 shows the parameter count of parameters of FITS with different settings on the ETTm1 & 2 datasets. Tab. 6 and Tab.7 show the corresponding results on ETTm1 and ETTm2 datasets with different settings. Note that FITS constantly achieves SOTA performance on the ETTm2 dataset with under 10k parameters.

Table 5: The number of parameters under different settings on ETTm1 & ETTm2 dataset.

| Horizon | COF/nth Harmonic | Look-back Window | | | |
|---|---|---|---|---|---|
| | | **90** | **180** | **360** | **720** |
| 96 | **4** | 420 | 513 | 621 | 1330 |
| | **6** | 561 | 759 | 1015 | 2444 |
| | **8** | 703 | 1053 | 1505 | 3835 |
| | **10** | 861 | 1426 | 2050 | 5609 |
| | **12** | 1035 | 1820 | 2726 | 7636 |
| 192 | **4** | 645 | 703 | 759 | 1505 |
| | **6** | 850 | 1035 | 1218 | 2726 |
| | **8** | 1064 | 1431 | 1820 | 4307 |
| | **10** | 1302 | 1922 | 2501 | 6248 |
| | **12** | 1564 | 2450 | 3290 | 8549 |
| 336 | **4** | 990 | 969 | 966 | 1715 |
| | **6** | 1275 | 1449 | 1566 | 3149 |
| | **8** | 1615 | 1998 | 2275 | 5015 |
| | **10** | 1974 | 2666 | 3157 | 7242 |
| | **12** | 2392 | 3395 | 4136 | 9960 |
| 720 | **4** | 1890 | 1710 | 1518 | 2380 |
| | **6** | 2448 | 2530 | 2436 | 4324 |
| | **8** | 3078 | 3510 | 3570 | 6844 |
| | **10** | 3780 | 4650 | 4920 | 9940 |
| | **12** | 4554 | 5950 | 6486 | 13612 |

Table 6: The results on the ETTm1 dataset. Values are visualized with a green background, where darker background indicates worse performance. The top-5 best results are highlighted with a red background, and the absolute best result is highlighted with **red bold** font. **F** represents supervision on the forecasting task, while **B+F** represents supervision on backcasting and forecasting tasks.

| Horizon | Input length COF/nth Harmonic | 90 | | 180 | | 360 | | 720 | |
|---|---|---|---|---|---|---|---|---|---|
| | | **F** | **B+F** | **F** | **B+F** | **F** | **B+F** | **F** | **B+F** |
| 96 | **4** | 0.36539 | 0.368287 | 0.315333 | 0.314558 | 0.311276 | 0.310786 | 0.323238 | 0.321785 |
| | **6** | 0.366902 | 0.36644 | 0.316294 | 0.314923 | 0.308139 | 0.307256 | 0.320556 | 0.318702 |
| | **8** | 0.363857 | 0.364832 | 0.314383 | 0.314648 | **0.305866** | 0.306163 | 0.313537 | 0.315031 |
| | **10** | 0.365007 | 0.366312 | 0.313453 | 0.312554 | 0.30812 | 0.308093 | 0.313483 | 0.315461 |
| | **12** | 0.362493 | 0.364372 | 0.314225 | 0.314401 | 0.306075 | 0.306781 | 0.313896 | 0.316464 |
| 192 | **4** | 0.402017 | 0.411609 | 0.350637 | 0.351265 | 0.344014 | 0.342416 | 0.347725 | 0.3501 |
| | **6** | 0.402813 | 0.413139 | 0.349372 | 0.352178 | 0.339382 | 0.34013 | 0.344214 | 0.345845 |
| | **8** | 0.403378 | 0.408561 | 0.350732 | 0.35014 | 0.340666 | 0.339582 | 0.341009 | 0.341524 |
| | **10** | 0.402122 | 0.409548 | 0.35038 | 0.351084 | 0.340434 | 0.339451 | 0.341734 | 0.343237 |
| | **12** | 0.404392 | 0.416905 | 0.348248 | 0.350706 | 0.339771 | **0.339245** | 0.342307 | 0.341273 |
| 336 | **4** | 0.457025 | 0.626306 | 0.38627 | 0.498018 | 0.376918 | 0.378298 | 0.373669 | 0.375122 |
| | **6** | 0.453296 | 0.604673 | 0.387365 | 0.444529 | 0.37518 | 0.374951 | 0.372264 | 0.371787 |
| | **8** | 0.478034 | 0.631451 | 0.387241 | 0.440868 | 0.373784 | 0.374112 | 0.368906 | 0.37 |
| | **10** | 0.5171 | 0.629327 | 0.394267 | 0.468663 | 0.37465 | 0.373348 | **0.367833** | 0.369997 |
| | **12** | 0.443713 | 0.63693 | 0.386978 | 0.412943 | 0.373993 | 0.374413 | 0.370057 | 0.368717 |
| 720 | **4** | 0.593805 | 0.664672 | 0.460092 | 0.634543 | 0.431741 | 0.459255 | 0.422784 | 0.422793 |
| | **6** | 0.599244 | 0.677897 | 0.47532 | 0.665818 | 0.430778 | 0.456598 | 0.422234 | 0.423324 |
| | **8** | 0.620804 | 0.70239 | 0.462694 | 0.620843 | 0.433994 | 0.474499 | **0.418575** | 0.42085 |
| | **10** | 0.564101 | 0.723161 | 0.459982 | 0.653963 | 0.432989 | 0.470532 | 0.418746 | 0.419788 |
| | **12** | 0.604411 | 0.730127 | 0.542039 | 0.625938 | 0.432008 | 0.485034 | 0.420789 | 0.424112 |

Table 7: The results on the ETTm2 dataset. Values are visualized with a green background, where darker background indicates worse performance. The top-5 best results are highlighted with a red background, and the absolute best result is highlighted with **red bold** font. **F** represents supervision on the forecasting task, while **B+F** represents supervision on backcasting and forecasting tasks.

| Horizon | COF/nth Harmonic | 90 F | 90 B+F | 180 F | 180 B+F | 360 F | 360 B+F | 720 F | 720 B+F |
|---|---|---|---|---|---|---|---|---|---|
| 96 | 4 | 0.189949 | 0.187104 | 0.175861 | 0.175653 | 0.168331 | 0.167802 | 0.167627 | 0.168365 |
| | 6 | 0.187921 | 0.187752 | 0.175081 | 0.174664 | 0.167421 | 0.166934 | 0.165331 | 0.166699 |
| | 8 | 0.18755 | 0.186862 | 0.174284 | 0.174376 | 0.167456 | 0.166398 | 0.16545 | 0.165945 |
| | 10 | 0.186856 | 0.187068 | 0.174272 | 0.174055 | 0.166025 | 0.166027 | 0.164797 | 0.165304 |
| | 12 | 0.188115 | 0.187032 | 0.174164 | 0.17395 | 0.166229 | 0.16578 | **0.16419** | 0.164371 |
| 192 | 4 | 0.250291 | 0.250258 | 0.235167 | 0.235682 | 0.222561 | 0.221472 | 0.221164 | 0.2204 |
| | 6 | 0.25162 | 0.251188 | 0.234177 | 0.234117 | 0.222139 | 0.221428 | 0.219001 | 0.218901 |
| | 8 | 0.250965 | 0.252477 | 0.234083 | 0.234356 | 0.221141 | 0.221143 | 0.219023 | 0.21849 |
| | 10 | 0.251273 | 0.252961 | 0.233744 | 0.23406 | 0.220952 | 0.220169 | 0.218367 | 0.217286 |
| | 12 | 0.250632 | 0.250457 | 0.233587 | 0.234173 | 0.22108 | 0.220789 | 0.217687 | **0.217022** |
| 336 | 4 | 0.311742 | 0.314966 | 0.289996 | 0.28909 | 0.27523 | 0.275501 | 0.272427 | 0.271816 |
| | 6 | 0.311689 | 0.315261 | 0.289143 | 0.289707 | 0.275103 | 0.275547 | 0.271085 | 0.270759 |
| | 8 | 0.317793 | 0.319121 | 0.288993 | 0.294166 | 0.274604 | 0.274989 | 0.270647 | 0.270937 |
| | 10 | 0.311076 | 0.326517 | 0.289358 | 0.290076 | 0.274078 | 0.274672 | 0.270995 | 0.270266 |
| | 12 | 0.311036 | 0.323915 | 0.288891 | 0.291164 | 0.273783 | 0.274388 | 0.269596 | **0.269525** |
| 720 | 4 | 0.41408 | 0.420778 | 0.385617 | 0.407686 | 0.365705 | 0.367553 | 0.350079 | 0.349886 |
| | 6 | 0.412397 | 0.423905 | 0.385204 | 0.410507 | 0.36524 | 0.369753 | 0.349508 | 0.348787 |
| | 8 | 0.418551 | 0.43163 | 0.386254 | 0.406818 | 0.365354 | 0.371821 | 0.349908 | 0.349498 |
| | 10 | 0.415603 | 0.427358 | 0.387376 | 0.411223 | 0.364901 | 0.371391 | 0.348837 | 0.347984 |
| | 12 | 0.420396 | 0.43113 | 0.394693 | 0.404091 | 0.365673 | 0.370805 | 0.348593 | **0.347862** |

## C.2 Traffic

Tab. 8 shows the parameter count of parameters of FITS with different settings on the Traffic dataset. Tab. 9shows the result on the Traffic dataset with different settings correspondingly. The traffic dataset has a very large amount of channels, making many models need many parameters to model the temporal information. FITS only needs 50k parameters to achieve comparable performance.

Table 8: The number of parameters under different settings on Traffic dataset.

| Horizon | COF/nth Harmonic | Look-back Window 90 | 180 | 360 | 720 |
|---|---|---|---|---|---|
| 96 | 3 | 1035 | 1820 | 4307 | 12064 |
| | 4 | 1431 | 2752 | 6975 | 20385 |
| | 5 | 1922 | 3876 | 10374 | 31042 |
| 192 | 3 | 1564 | 2450 | 5192 | 13520 |
| | 4 | 2187 | 3698 | 8475 | 22815 |
| | 5 | 2914 | 5253 | 12558 | 34694 |
| 336 | 3 | 2392 | 3395 | 6608 | 15704 |
| | 4 | 3321 | 5160 | 10725 | 26460 |
| | 5 | 4402 | 7293 | 15834 | 40006 |
| 720 | 3 | 4554 | 5950 | 10266 | 21424 |
| | 4 | 6318 | 9030 | 16650 | 36180 |
| | 5 | 8370 | 12750 | 24570 | 54780 |

Table 9: The results on the Traffic dataset. Values are visualized with a green background, where darker background indicates worse performance. The top-5 best results are highlighted with a red background, and the absolute best result is highlighted with **red bold** font. **F** represents supervision on the forecasting task, while **B+F** represents supervision on backcasting and forecasting tasks.

| Horizon | COF/nth Harmonic | 90 F | 90 B+F | 180 F | 180 B+F | 360 F | 360 B+F | 720 F | 720 B+F |
|---|---|---|---|---|---|---|---|---|---|
| 96 | 3 | 0.694065 | 0.694425 | 0.474606 | 0.475881 | 0.455815 | 0.457292 | 0.436317 | 0.436616 |
| | 4 | 0.690741 | 0.691064 | 0.462886 | 0.463642 | 0.434575 | 0.434842 | 0.414185 | 0.415293 |
| | 5 | 0.688774 | 0.691499 | 0.459929 | 0.459652 | 0.423814 | 0.422843 | **0.401225** | 0.403405 |
| 192 | 3 | 0.627212 | 0.636434 | 0.481686 | 0.485085 | 0.463516 | 0.46417 | 0.442661 | 0.443547 |
| | 4 | 0.625307 | 0.649024 | 0.470148 | 0.483849 | 0.439995 | 0.440732 | 0.4198 | 0.419938 |
| | 5 | 0.623088 | 0.636091 | 0.466362 | 0.478839 | 0.429684 | 0.4296 | **0.407131** | 0.408353 |
| 336 | 3 | 0.635301 | 0.662283 | 0.4962 | 0.510793 | 0.47309 | 0.476491 | 0.454243 | 0.456989 |
| | 4 | 0.63295 | 0.656833 | 0.484066 | 0.50553 | 0.451054 | 0.454847 | 0.432025 | 0.433721 |
| | 5 | 0.631095 | 0.670716 | 0.480058 | 0.512274 | 0.439686 | 0.444552 | 0.420825 | 0.423244 |
| 720 | 3 | 0.685472 | 0.732168 | 0.529004 | 0.606921 | 0.500635 | 0.587891 | 0.488116 | 0.489934 |
| | 4 | 0.684401 | 0.752384 | 0.515154 | 0.617825 | 0.481284 | 0.58569 | 0.468166 | 0.469335 |
| | 5 | 0.688761 | 0.752565 | 0.523269 | 0.628914 | 0.472644 | 0.583838 | **0.456807** | 0.460696 |

Table 10: The number of parameters under different settings on Weather dataset.

| Horizon | COF/nth Harmonic | Look-back Window | | | |
|---|---|---|---|---|---|
| | | 90 | 180 | 360 | 720 |
| 96 | 3 | 364 | 408 | 480 | 899 |
| | 4 | 420 | 513 | 621 | 1330 |
| | 5 | 496 | 630 | 806 | 1845 |
| | 8 | 703 | 1053 | 1505 | 3835 |
| 192 | 3 | 560 | 561 | 580 | 1015 |
| | 4 | 645 | 703 | 759 | 1505 |
| | 5 | 752 | 861 | 988 | 2050 |
| | 8 | 1064 | 1431 | 1820 | 4307 |
| 336 | 3 | 854 | 765 | 720 | 1189 |
| | 4 | 990 | 969 | 966 | 1715 |
| | 5 | 1136 | 1197 | 1248 | 2378 |
| | 8 | 1615 | 1998 | 2275 | 5015 |
| 720 | 3 | 1638 | 1360 | 1140 | 1624 |
| | 4 | 1890 | 1710 | 1518 | 2380 |
| | 5 | 2160 | 2100 | 1950 | 3280 |
| | 8 | 3078 | 3510 | 3570 | 6844 |

Table 11: The results on the Weather dataset. Values are visualized with a green background, where darker background indicates worse performance. The top-5 best results are highlighted with a red background, and the absolute best result is highlighted with **red bold** font. **F** represents supervision on the forecasting task, while **B+F** represents supervision on backcasting and forecasting tasks.

| Horizon | COF/nth Harmonic | 90 | | 180 | | 360 | | 720 | |
|---|---|---|---|---|---|---|---|---|---|
| | | F | B+F | F | B+F | F | B+F | F | B+F |
| 96 | 3 | 0.197956 | 0.198834 | 0.190438 | 0.190583 | 0.177569 | 0.176701 | 0.174107 | 0.173947 |
| | 4 | 0.19808 | 0.198548 | 0.190979 | 0.19016 | 0.176951 | 0.175991 | 0.172446 | 0.172651 |
| | 5 | 0.198305 | 0.197615 | 0.189992 | 0.190143 | 0.175894 | 0.176468 | 0.172261 | 0.173201 |
| | 8 | 0.197515 | 0.197714 | 0.189467 | 0.190344 | 0.175324 | 0.174741 | 0.171744 | **0.170606** |
| 192 | 3 | 0.243689 | 0.244304 | 0.234231 | 0.233943 | 0.219906 | 0.219789 | 0.216264 | 0.21634 |
| | 4 | 0.243442 | 0.244047 | 0.233548 | 0.233765 | 0.219619 | 0.2193 | 0.215846 | 0.215159 |
| | 5 | 0.244325 | 0.244027 | 0.232503 | 0.233373 | 0.218952 | 0.219246 | 0.215042 | 0.214364 |
| | 8 | 0.243439 | 0.244354 | 0.233635 | 0.232369 | 0.218155 | 0.218985 | 0.214377 | **0.214347** |
| 336 | 3 | 0.296318 | 0.299125 | 0.281715 | 0.284938 | 0.266293 | 0.266286 | 0.260424 | 0.259934 |
| | 4 | 0.295563 | 0.298132 | 0.281794 | 0.284505 | 0.266475 | 0.266438 | 0.260124 | 0.259734 |
| | 5 | 0.295225 | 0.299156 | 0.281916 | 0.287063 | 0.265812 | 0.26592 | **0.259221** | 0.259553 |
| | 8 | 0.295462 | 0.301229 | 0.281217 | 0.288032 | 0.26527 | 0.265357 | 0.259368 | 0.259352 |
| 720 | 3 | 0.368714 | 0.373197 | 0.353147 | 0.358274 | 0.333046 | 0.334346 | 0.321251 | 0.32157 |
| | 4 | 0.369893 | 0.374172 | 0.352436 | 0.355783 | 0.332602 | 0.335239 | 0.32068 | 0.322293 |
| | 5 | 0.3691 | 0.373669 | 0.352941 | 0.358575 | 0.332862 | 0.336928 | 0.321146 | 0.321193 |
| | 8 | 0.368921 | 0.376838 | 0.353168 | 0.361575 | 0.33276 | 0.334887 | **0.32057** | 0.321736 |

## C.3 Weather

Tab. 10 shows the parameter count of parameters of FITS with different settings on the Weather dataset. Tab. 9shows the result on the Traffic dataset with different settings correspondingly. Note that we achieve the result in the main table by setting the COF as 75 and the look-back window as 700.

## C.4 Electricity

Tab. 12 shows the parameter count of parameters of FITS with different settings on the Electricity dataset. Tab. 13 shows the result on the Electricity dataset with different settings correspondingly. We find that the Electricity dataset is sensitive to the COF. This is because this dataset shows significant multi-periodicity, which requires capturing high-frequency components. Otherwise, FITS will not learn such information.

Table 12: The number of parameters under different settings on Electricity dataset.

| Horizon | COF/nth Harmonic | Look-back Window | | | |
|---|---|---|---|---|---|
| | | **90** | **180** | **360** | **720** |
| 96 | 2 | 703 | 1053 | 2279 | 5913 |
| | 3 | 1035 | 1820 | 4307 | 12064 |
| | 4 | 1431 | 2752 | 6975 | 20385 |
| | 5 | 1922 | 3876 | 10374 | 31042 |
| | 8 | 3698 | 8475 | 24186 | 75628 |
| 192 | 2 | 703 | 1053 | 2279 | 5913 |
| | 3 | 1035 | 1820 | 4307 | 12064 |
| | 4 | 1431 | 2752 | 6975 | 20385 |
| | 5 | 1922 | 3876 | 10374 | 31042 |
| | 8 | 3698 | 8475 | 24186 | 75628 |
| 336 | 2 | 1615 | 1998 | 3483 | 7665 |
| | 3 | 2392 | 3395 | 6608 | 15704 |
| | 4 | 3321 | 5160 | 10725 | 26460 |
| | 5 | 4402 | 7293 | 15834 | 40006 |
| | 8 | 8514 | 15900 | 36974 | 97902 |
| 720 | 2 | 3078 | 3510 | 5418 | 10512 |
| | 3 | 4554 | 5950 | 10266 | 21424 |
| | 4 | 6318 | 9030 | 16650 | 36180 |
| | 5 | 8370 | 12750 | 24570 | 54780 |
| | 8 | 16254 | 27750 | 57546 | 133644 |

Table 13: The results on the Electricity dataset. Values are visualized with a green background, where darker background indicates worse performance. The top-5 best results are highlighted with a red background, and the absolute best result is highlighted with **red bold** font. **F** represents supervision on the forecasting task, while **B+F** represents supervision on backcasting and forecasting tasks.

| | Input length | 90 | | 180 | | 360 | | 720 | |
|---|---|---|---|---|---|---|---|---|---|
| Horizon | COF/nth Harmonic | **F** | **B+F** | **F** | **B+F** | **F** | **B+F** | **F** | **B+F** |
| 96 | 2 | 0.219861 | 0.220638 | 0.180801 | 0.181306 | 0.187296 | 0.18764 | 0.182412 | 0.182459 |
| | 3 | 0.214897 | 0.220059 | 0.170256 | 0.170923 | 0.167465 | 0.167297 | 0.162607 | 0.16269 |
| | 4 | 0.211179 | 0.211584 | 0.164911 | 0.164888 | 0.156822 | 0.156631 | 0.151266 | 0.15207 |
| | 5 | 0.20964 | 0.210331 | 0.1614 | 0.162626 | 0.151409 | 0.15133 | 0.1457 | 0.146501 |
| | 8 | 0.205661 | 0.207206 | 0.155651 | 0.156382 | 0.142126 | 0.142532 | 0.139022 | **0.13841** |
| 192 | 2 | 0.216865 | 0.22451 | 0.195207 | 0.192494 | 0.200281 | 0.20018 | 0.195381 | 0.195814 |
| | 3 | 0.211918 | 0.224618 | 0.191438 | 0.182895 | 0.180623 | 0.180136 | 0.175342 | 0.175513 |
| | 4 | 0.210492 | 0.223837 | 0.179412 | 0.18544 | 0.169962 | 0.169827 | 0.164689 | 0.165129 |
| | 5 | 0.207801 | 0.217388 | 0.17504 | 0.177671 | 0.164243 | 0.165165 | 0.159636 | 0.160017 |
| | 8 | 0.205524 | 0.216681 | 0.169276 | 0.194963 | 0.155918 | 0.157119 | 0.154376 | **0.152842** |
| 336 | 2 | 0.232701 | 0.248004 | 0.213782 | 0.236475 | 0.2144 | 0.215592 | 0.209039 | 0.20925 |
| | 3 | 0.228432 | 0.253425 | 0.205334 | 0.230306 | 0.196429 | 0.195466 | 0.189955 | 0.189872 |
| | 4 | 0.223753 | 0.243039 | 0.207252 | 0.225273 | 0.18502 | 0.185789 | 0.180464 | 0.17935 |
| | 5 | 0.22162 | 0.252762 | 0.195054 | 0.23265 | 0.184309 | 0.181705 | 0.174788 | 0.175183 |
| | 8 | 0.220267 | 0.247537 | 0.190028 | 0.242119 | 0.171824 | 0.175581 | 0.166873 | **0.166276** |
| 720 | 2 | 0.279359 | 0.304686 | 0.25978 | 0.311533 | 0.255001 | 0.285431 | 0.249335 | 0.249288 |
| | 3 | 0.271019 | 0.303177 | 0.250443 | 0.319552 | 0.234653 | 0.266245 | 0.230563 | 0.23081 |
| | 4 | 0.272873 | 0.30215 | 0.267512 | 0.348516 | 0.228953 | 0.266807 | 0.219166 | 0.224155 |
| | 5 | 0.274384 | 0.308015 | 0.283174 | 0.346803 | 0.222556 | 0.259166 | 0.212817 | 0.215066 |
| | 8 | 0.274878 | 0.320414 | 0.271509 | 0.383154 | 0.221972 | 0.298277 | **0.205646** | 0.210626 |

## D Full Anomaly Detection Results

The full results with Accuracy, Precision, Recall, and F1-score are shown in Tab. 14. For better performance, we also conduct experiments only on the first channel of the SML dataset, denoted as (C0). We also trained FITS using only the analog channels of SWaT, denoted as (analog).

Table 14: Full results on five datasets.

| Datasets | Accuracy | Precision | Recall | F1-score |
|---|---|---|---|---|
| SMD | 99.92 | 99.9 | 100 | 99.95 |
| PSM | 94.43 | 97.2 | 90.43 | 93.69 |
| SWaT | 99.42 | 97.84 | 100 | 98.9 |
| SWaT(analog) | 97.81 | 91.74 | 100 | 95.69 |
| SMAP | 89.39 | 77.52 | 65.05 | 70.74 |
| MSL | 81.52 | 61.38 | 80.16 | 69.52 |
| MSL(C0) | 83.77 | 81.34 | 75.15 | 78.12 |

## E Datasets Visualization on Anomaly Detection

As shown in Fig. 2 and Fig. 3, most PSM and SMD datasets channels are analog values. Especially the PSM dataset shows great periodicity.

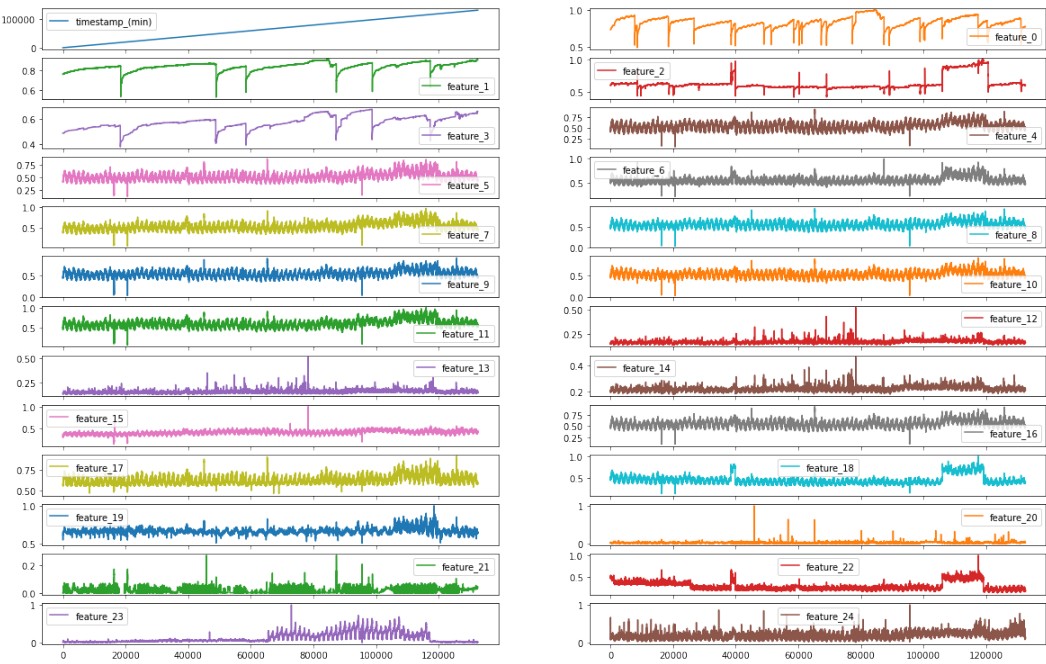

Figure 2: Waveform of PSM dataset.

While some channels in the SWaT dataset are binary event values, as shown in Fig. 4.

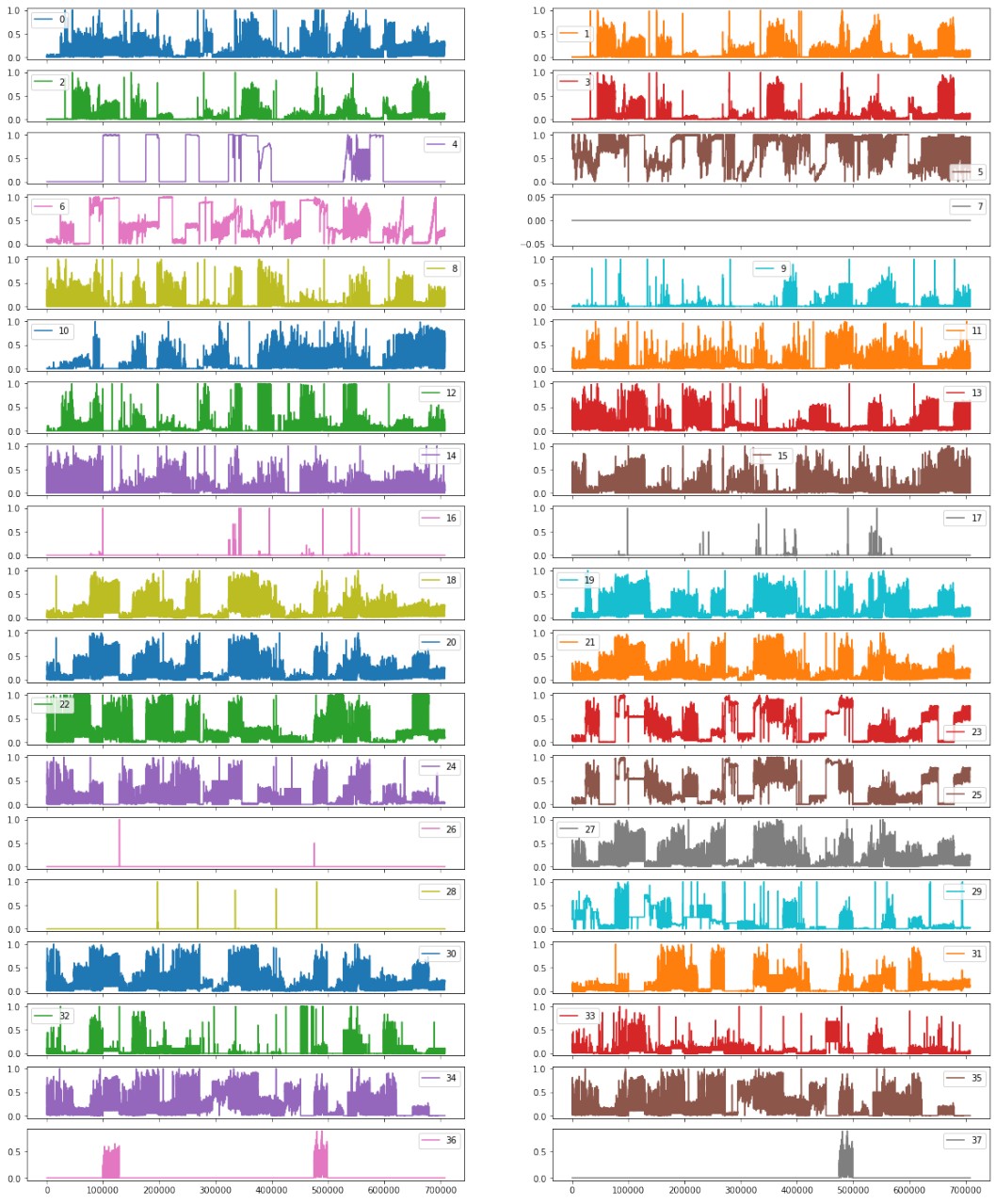

Figure 3: Waveform of SMD dataset.

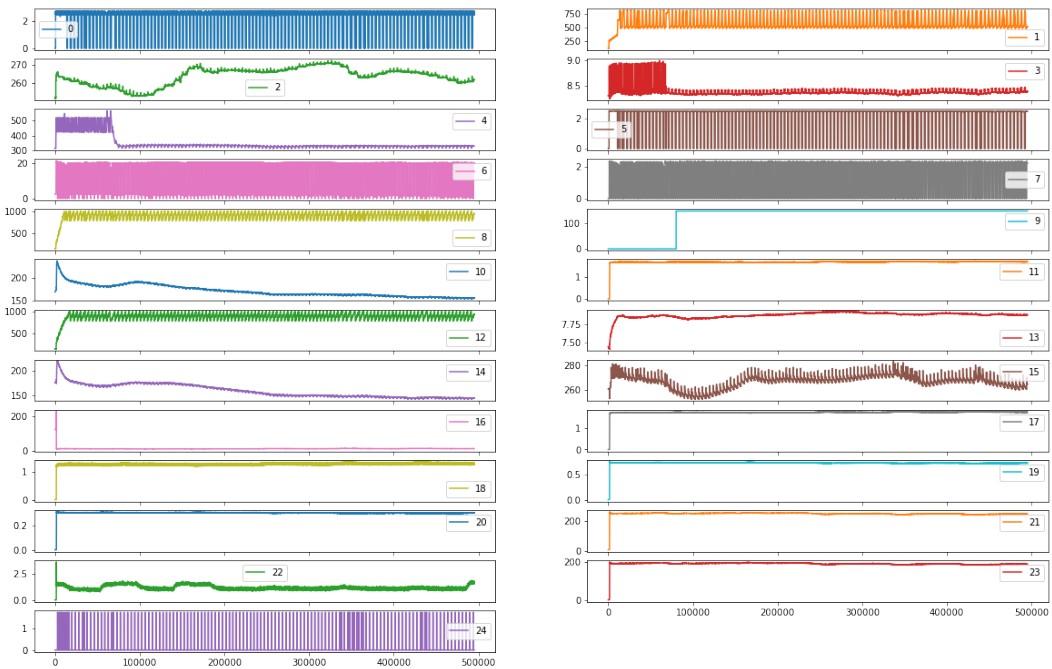

Figure 4: Waveform of SWAT dataset.

However, as shown in Fig. 5 and Fig. 6, for SMAP and MSL datasets, most channels are binary event values that are hard for FITS to learn frequency representation.

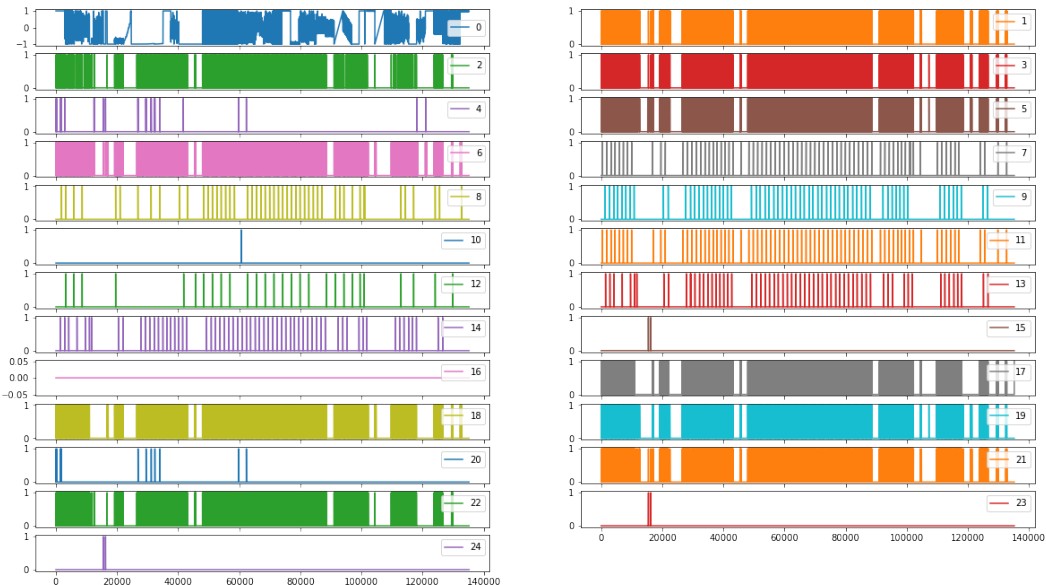

Figure 5: Waveform of SMAP dataset.

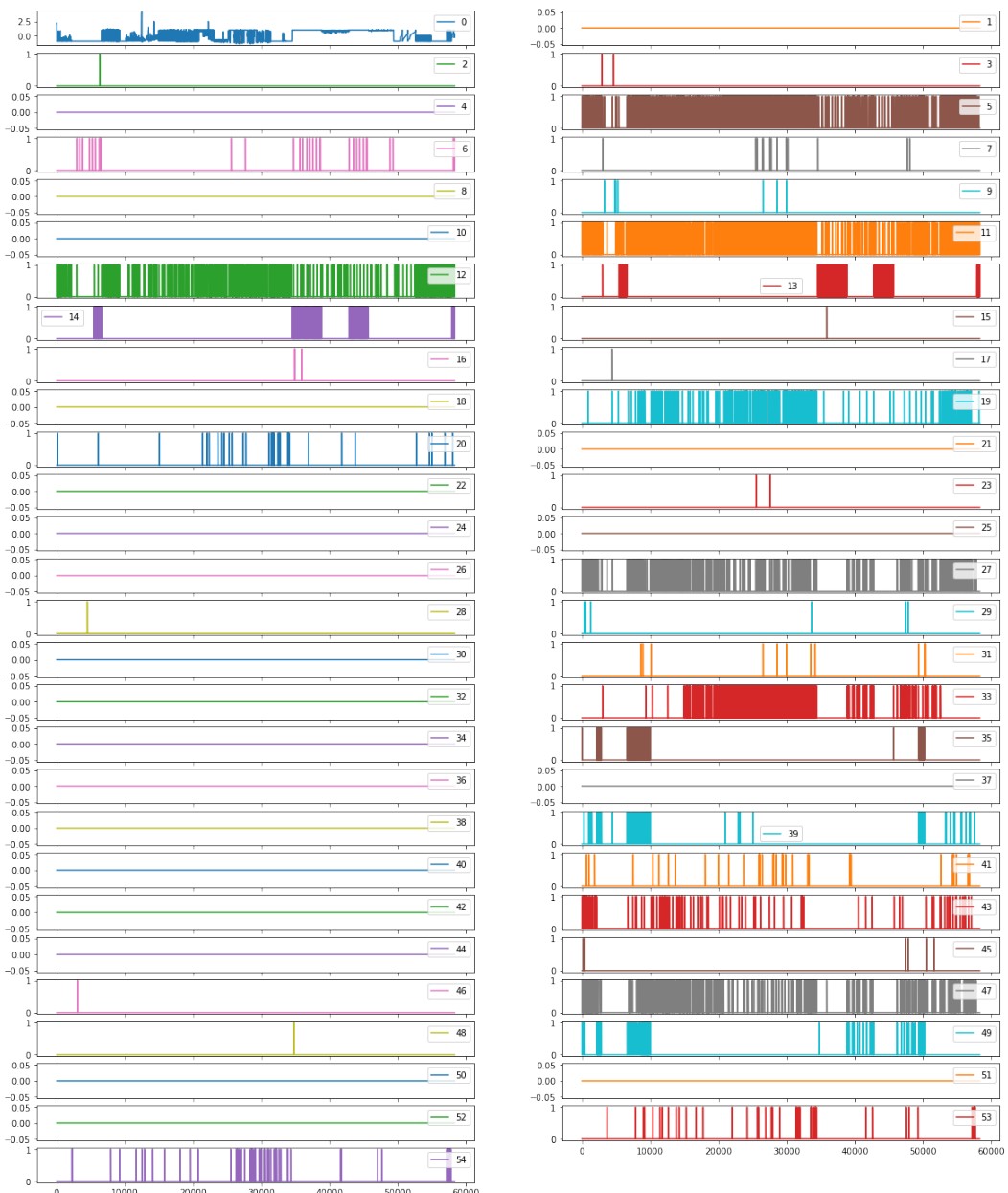

Figure 6: Waveform of MSL dataset.

# F    Parameter Counts for Anomaly Detection

We use a fixed sliding window of 200 and 400 for all the datasets and do not apply any frequency filter. The downsample rate is set as 4 for any dataset. Thus, the number of parameters is as Tab. 15.

Table 15: MACs and parameter count of FITS on Anomaly Detection task. We report the MACs on the SWaT dataset which has 55 channels.

| Window | Params | MACs |
|--------|--------|--------|
| 200 | 2600 | 137.5k |
| 400 | 10200 | 550k |