# OpenReview forum: "FITS: Modeling Time Series with 10k Parameters"
_NeurIPS.cc/2023/Conference — Submitted to NeurIPS 2023_

### Official Review · Reviewer_o3Wj · 2023-06-13

**Soundness:** 3 good
**Presentation:** 2 fair
**Contribution:** 2 fair
**Rating:** 5
**Confidence:** 3

**Summary:**

The authors introduce FITS, a lightweight model for time series analysis. Unlike existing models that process raw time-domain data directly, FITS operates on the principle of manipulating time series through interpolation in the complex frequency domain. By discarding high-frequency components that have a negligible impact on the time series data, FITS achieves performance comparable to state-of-the-art models for time series forecasting and anomaly detection tasks. Additionally, FITS has a remarkably compact size, consisting of only approximately 10k parameters. This lightweight model is mainly constituted by a simple Complex-valued Linear Layer functioning on frequency domain.

**Strengths:**

- About the presentation: Straight-forward to follow, simple and direct sentences.
- About the contribution: The proposed FITS framework is very light-weighted, since its learning parameters mainly coming from 1 dense layer. This contribution make its great application for actual real-world scenarios, applying on edge devices. The natural idea about *Low Pass Filter* is not new, but is used with good reasoning.
- About the experiment: Authors illustrate the effectivenesses of FITS with two main time series tasks: Forecasting and detecting anomaly. While these experiments are not extensive, it effectively support the contributions the authors claim: Comparative performance and much more lightweight compared to existing SOTAs.

**Weaknesses:**

- About the presentation: Figure 1 has some too dim colors and hard to see; Figure 2 is also quite small, the font size should be increased.
- About the contribution:
    - Details about input or output dimensions are not discussed (only discuss about the temporal axis). It is not clear whether the framework can be applied to multivariate series? How to choose cutoff frequency in case of multivariate series, when the harmonics are likely to be different over different variables?
    - While the different components constituting FITS are used with clear intentions, these techniques or algorithms (rFFT, RIN, low-pass filter, …) are not new and even well established.
    - The experiments for both time series and anomaly detection tasks suggest the framework has a great variance when evaluating on different datasets. In general, FITS cannot achieve SOTA performance in many scenarios, which may be unsuitable for performance-critical applications.

**Questions:**

- Are there any plans to enhance the visibility of the figure, perhaps by adjusting the colors, the font size, or using a different visualization technique?
- Can the proposed framework be extended to handle multivariate time series? If so, how do you choose the cutoff frequency when the harmonics are likely to differ across different variables?

**Limitations:**

Authors recognize following weaknesses:
- FITS struggles with binary-valued time series and time series with missing data.
- Binary-valued time series are better suited for time-domain modeling due to their compact raw data format.
- For time series with missing data, a two-step approach is suggested: apply simple time-domain imputation techniques before utilizing FITS for analysis.

The authors should consider amend the weaknesses mentioned above if possible or make some modifications to the manuscripts to rebuttal the comments.

---

> ### Author Rebuttal · Authors · 2023-08-07
>
> We sincerely apologize for the inconvenience caused by the color scheme of the figures. In the revised version, we will change the color scheme to enhance visibility and ensure a better viewing experience for readers.
>
> - W2.1, Q2: Can the proposed framework be extended to handle multivariate time series? If so, how do you choose the cutoff frequency when the harmonics are likely to differ across different variables?
>
>
> ***Certainly, FITS is adept at handling multivariate time series data.*** *Our experimentation, as highlighted in Table 1 of the original manuscript, was conducted on multivariate datasets, thereby showcasing FITS' competence in managing multi-channel data. Detailed dataset dimensions are provided in Table 1.*
>
> *We choose the harmonic cutoff by observing the amplitude spectrum and selecting the largest harmonic that appears across the dataset as a cutoff frequency. However, our experiment shows that the performance boost introduced by increasing the cutoff frequency is minor after the cutoff frequency is larger than the second/third order harmonic.*
>
> *Concerning datasets with different base frequencies,* *our observation suggests that the likelihood of inter-channel correlation is minimal. It's important to note that individual cutoff frequencies can be specified for each channel, providing the flexibility to tailor the choice according to specific data characteristics.*
>
> ***For detailed discussions, please refer to the common response 'About Multivariate' section.***
>
> - W2.2: About the novelty:
>
> *We acknowledge that the specific blocks and algorithms are not our unique contributions. The LPF and RIN are well-established techniques and Complex-valued neural networks can even be dated back to the 19th century. But as you mention in the Strength, we select these techniques with good reasoning.*
>
> ***Our primary innovation lies in reformulating forecasting and reconstruction tasks as frequency domain interpolations.** We introduce an effective approach to leverage frequency domain information using complex-valued neural networks.*
>
> *Finally, FITS represents a significant contribution as an exceptionally lightweight TSA model suitable for deployment on edge devices with comparable or even superior performance to the SOTA models. This counterintuitively demonstrates that compact models can achieve comparable or even superior performance when compared to larger models.*
>
> - W2.3: About AD performance:
>
> *Indeed, FITS may not be the optimal choice for performance-critical applications. However, it is designed as a lightweight algorithm tailored for edge devices with limited resources (e.g., smart sensors), bringing forth distinct advantages. FITS demands minimal memory and computational resources, and notably, it demonstrates **sub-millisecond level inference time.** This temporal efficiency is inconsequential when compared to the time taken for inferencing a large model or even the communication. These characteristics render FITS exceptionally well-suited for the **swift detection of critical errors and rapid response**. By integrating FITS as a coarse-grained filter, followed by a specialized AD algorithm for finer-grained detection, the overall system achieves robustness and responsiveness, addressing both severe and nuanced anomalies effectively.*

---

> > ### Comment · Reviewer_o3Wj · 2023-08-18
> > **Extra comment on authors' rebuttal**
> >
> > First, I want to thank the authors for responding.
> > However, here are some extra comments about the explanations:
> > - W2.1: I understand that for multivariate cases, multiple base frequencies are selected but with the introduce of an extra assumption about uncorrelated channels (variables). This assumption (which is not held for all datasets) should be put in the revised manuscript, together with how FITS handle multivariate data.
> > - W2.2: Your answer do not bring any new information.
> > With these, I will keep the score as it is.

---

> > > ### Author Response · Authors · 2023-08-18
> > >
> > > We sincerely thank the reviewer for the responses to our rebuttal, and we would like to further clarify these two questions.
> > >
> > > - W2.1 I understand that for multivariate cases, multiple base frequencies are selected but with the introduce of an extra assumption about uncorrelated channels (variables). This assumption (which is not held for all datasets) should be put in the revised manuscript, together with how FITS handle multivariate data.
> > >
> > > Thank you for the suggestion and we shall clarify it in the revised version.
> > >
> > > At the same time, it seems that our earlier response is not crystal clear. As mentioned in the common response at the top rebuttal section, FITS handles multivariate data by weight sharing. In practice, channels often share a common base frequency when originating from the same physical system. For instance, signals from electrical appliances commonly have a base frequency of 50/60Hz, while traffic flow across a city follows a daily base frequency. Most of the datasets used in our experiments possess such attributes, and hence we simply apply weight sharing strategy for them. Such an approach balances performance and efficiency. For datasets that indeed contain channels with different base frequencies, we can cluster those channels according to the base frequency and train an individual FITS model for each cluster. We shall elaborate on the above in the revised version.
> > >
> > > - W2.2 Your answer do not bring any new information.
> > >
> > > We respect the reviewer's opinion and agree that the various components used in our FITS model are known in the literature. Our main contribution is the proposed overall framework, and it is well summarized by the reviewer: "The proposed FITS framework is very light-weighted, since its learning parameters mainly coming from 1 dense layer. This contribution make its great application for actual real-world scenarios, applying on edge devices.".

---

> > > > ### Comment · Reviewer_o3Wj · 2023-08-19
> > > > **Comment on authors' replies**
> > > >
> > > > Thanks for further clarification.
> > > >
> > > > W2.1. With your clarification, I do think that revision should be quite major.
> > > > You should dedicate a section for multivariate handling, together with empirical prove (e.g. for datasets with correlated channels and uncorrelated channels) if possible.
> > > > You can change main manuscript and provide details in the supplementary material.
> > > >
> > > > With your realization, I increase the score a little bit.

---

> > > > > ### Author Response · Authors · 2023-08-19
> > > > >
> > > > > We sincerely thank the reviewer for the positive rating of our submission. We shall follow the suggestions, i.e., update the revised version with a dedicated section on "multivariate time series handling" and add empirical results under various channel correlations.

---

### Official Review · Reviewer_dtUP · 2023-06-27

**Soundness:** 3 good
**Presentation:** 3 good
**Contribution:** 2 fair
**Rating:** 6
**Confidence:** 5

**Summary:**

This paper builds a model for time series learning in the frequency domain. The key idea is to discard high-frequency components to reduce the model size. This leads to a simple model under 10k parameters, 50x smaller than DLinear. Experimental results suggest the model can achieve comparable performance to the state-of-the-art methods on long-term forecasting tasks and several anomaly detection tasks.

**Strengths:**

1. Transforming time series into a frequency domain and then training a model is an interesting idea.
2. The model is extremely small, even 50X smaller than DLinear, which is already very small.
3. The experiments on long-term forecasting are strong.

**Weaknesses:**

1. It is unclear whether the approach has practical value. DLinear could be already sufficiently small to fit in a normal edge device. Thus, it is not persuading to have an even smaller model.
2. Training efficiency is not reported.
3. The experiments on anomaly detection are not strong. Firstly, all the baselines are neural networks. However, traditional methods like OCSVM and IForest can have strong performance [1]. Secondly, the existing anomaly detection datasets could be flawed. Thus, I am particularly interested to know how the proposed method performs in the synthetic data provided in [1]. The pattern-wise outliers in [1] are synthesized by modifying the sinusoidal waves. Thus, the proposed method seems to well align with the design of this dataset. I am curious how it performs on this dataset.

[1] Revisiting Time Series Outlier Detection: Definitions and Benchmarks

**Questions:**

1. How does the proposed method perform on the synthetic anomaly dataset [1]?
2. Can the proposed method outperform classical anomaly detection methods, such as OCSVM and IForest?
3. Why do we need such a small model, as DLinear may already be small enough for most of the edge devices?

**Limitations:**

Yes

---

> ### Author Rebuttal · Authors · 2023-08-04
>
> Thanks for your insightful questions.
>
> - W1&Q3: Why do we need such a small model, as DLinear may already be small enough for most of the edge devices?:
>
> A: *We acknowledge that DLinear is already a compact model. However, for edge devices powered by **MCU (Microcontroller Units), e.g., smart sensors**, it might only have limited onboard storage in the **sub-MB** range. For instance, commonly used STM32 Series devices have just 192KB flash memory, making it infeasible to accommodate DLinear's parameter size. Controllers for smart sensors, such as the 8051 and ATmega32, have even more restricted memory, with capacities of 4KB and 32KB, respectively. In such scenarios, FITS' model size in the **KB range** presents a more viable option, delivering comparable performance.*
>
> |MCU|ROM/Flash Memory|
> |:--|:--|
> |805x| 4KB / 8KB|
> |ATmega328P (Arduino UNO)|32 KB|
> |STM32 Series|Up to 192KB|
> |ESP32 Series|4MB / 8MB|
>
> *For the devices that are deployed off the grid, energy efficiency is also a critical factor. FITS requires less calculation and hence can run with extended battery life.* We will address this importance in our revised version.
>
> *More importantly, FITS is capable of handling anomaly detection tasks, which is not the case for DLinear, as shown in the results of  TimesNet [1]. As discussed in the **'About Anomaly Detection' section of the common response**, FITS can capture critical errors swiftly thanks to its **sub-millisecond level inference time**. Such time consumption is even negligible compared with the communication latency. Edge devices can shut down the defective machine immediately once the fatal anomaly is detected, preventing the system from further damage. Furthermore, FITS can be integrated with other AD algorithms to achieve better performance, as described in the common response.*
>
> - W3&Q1&Q2: About Baseline and Dataset of AD task:
>
> A: *We show the comparison with the mentioned OCSVM and IForest in the common response. And find FITS **delivers superior performance on all five datasets**. Please refer to the **common response** for a detailed analysis.*
>
> *The mentioned dataset is really aligned with our FITS. We will report the performance on them in the revised version.*
>
> *However, we would like to emphasize that conducting AD is for demonstrating the use cases of FITS instead of claiming its superior performance.*
>
> - W2: About training efficiency:
>
> A: *FITS has very **high training efficiency**. With a single NVIDIA Titan Xp, FITS can finish training within 5 minutes on ETT datasets and 30 minutes on the Traffic dataset (which has 862 channels). Interestingly, on a super lightweight model such as FITS and DLinear, we find that the **bottleneck is the loss computation on CPU instead of the model itself**. Considering the compact size of FITS, the memory footprint is also very minor.*
>
> *For detailed analysis, please refer to the **'About Efficiency' section** of the common response G2.*
>
>
> [1] Haixu Wu, Tengge Hu, Yong Liu, Hang Zhou, Jianmin Wang, and Mingsheng Long. Timesnet: Temporal 2d-variation modeling for general time series analysis. In International Conference on Learning Representations, 2023.

---

> > ### Comment · Reviewer_dtUP · 2023-08-16
> > **Thank you for the reponse**
> >
> > Most of my concerns have been addressed, so I will increase my score. The authors are encouraged to do some analysis using the synthetic data (not only anomaly detection but it could also be used in forecasting tasks). This can help readers better understand how the algorithm works, and the pros/cons.

---

> > > ### Author Response · Authors · 2023-08-18
> > >
> > > We thank the reviewer for the recognition of our work.
> > >
> > > Following your suggestion, we conduct experiments on the aforementioned synthetic dataset, and the results are shown in the following table. We generate the synthetic dataset using the script provided in the benchmark with the default setting, i.e., 5% outlier on each channel with different outlier types. We generate 4000 time-steps as our dataset, in which we take 2500 for training and the rest 1500 for testing. For our FITS model, we use four different reconstruction windows, labeled as FITS-win{xxx}. We compare with the results retrieved from Table 17 of the original paper [1].
> > >
> > > |Model|Precision|Recall|F1-score|
> > > |---|---|---|---|
> > > |FITS-win24|1|1|1|
> > > |FITS-win50|1|1|1|
> > > |FITS-win100|1|0.9993|0.9996|
> > > |FITS-win400|1|0.9991|0.9995|
> > > |AR|0.59|0.77|0.64|
> > > |GBRT|0.47|0.56|0.51|
> > > |LSTM-RNN|0.22|0.26|0.24|
> > > |IForest|0.48|0.57|0.52|
> > > |OCSVM|0.62|0.74|0.67|
> > > |AutoEncoder|0.20|0.24|0.22|
> > > |GAN|0.15|0.15|0.15|
> > >
> > > The table clearly demonstrates FITS' superior performance compared to other models for this synthetic dataset. We attribute it to the setting of this synthetic dataset, which is constructed from a sinusoidal wave with a single frequency, augmented by added anomaly patterns. These patterns can be challenging to identify in the time domain. However, FITS excels in capturing these features in the frequency domain, allowing it to easily detect anomalies that introduce unexpected frequency components. For instance, consider a 16-point segment of a sinusoidal wave with a period of 8. The frequency representation should appear as [0, 0, 1+j, 0, 0, 0, 0, 0, 0]. After downsampling with a DSR of 4, the resulting 8-point segment exhibits a frequency representation of [0, 0, (1+j)*0.25, 0, 0]. FITS learns to reconstruct the original segment by scaling the frequency component at frequency 2 by 4 and zero-padding the frequency representation to a length of 9. However, anomalies introduce unexpected frequency components, which FITS is not trained to handle, leading to compromised reconstruction.
> > >
> > > Furthermore, we appreciate the intriguing suggestion to conduct forecasting experiments on synthetic datasets. Such experiments facilitate the behavior of frequency interpolation to be interpretable.
> > >
> > > Specifically, we conduct a frequency response test for FITS with a synthetic dataset that combines sinusoidal waves of four periods, i.e., 120, 60, 30, 24 (base, 2nd harmonic, 4th harmonic, and 5th harmonic). We train FITS with a cut_off frequency of 20, a look-back window of 240, and a forecasting horizon of 120. Our findings are as follows:
> > >
> > > - FITS can fit the dataset with ****zero loss**** (1e-9 on training/testing set). Instead of learning the complex temporal patterns, FITS is designed to learn on the frequency domain which can project the complex temporal patterns as four frequency components. All FITS needs to learn is linear projecting these four components to 4 different positions, e.g., repositioning frequency [2, 4, 8, 10] to frequency [3, 6, 12, 15].
> > > - FITS capably accommodates various combinations of frequencies present in the training set, thanks to the inherent independence of each frequency component in the frequency domain.
> > > - However, when faced with an unseen frequency during training, FITS yields very bad results since it hasn't been trained to project this particular frequency.
> > >
> > > We have provided the results and code within a Jupyter notebook titled "Interpretability.ipynb" in our anonymous code repository. This notebook contains additional visualizations and detailed results. We shall design more complex synthetic datasets to further demonstrate the performance and interpretability of our FITS model.
> > >
> > > Thanks again for your constructive suggestions!

---

> > > > ### Comment · Reviewer_dtUP · 2023-08-20
> > > > **Thank you for the further reponse**
> > > >
> > > > All my concerns are cleared. The authors are encouraged to provide more analysis and discussions with synthetic data to help readers better understand how the method behaves under different situations. I will increase my score further.

---

### Official Review · Reviewer_pZTr · 2023-07-02

**Soundness:** 3 good
**Presentation:** 3 good
**Contribution:** 3 good
**Rating:** 3
**Confidence:** 4

**Summary:**

This paper proposed a novel method on time-series forecasting and anomaly detection. The solution relies on the frequency-domain feature of the given time series and proposes to use a linear model to interpolate the data in the frequency domain. After that, the model utilizes the inverse FFT operation to transform the frequency domain data into time domain. The interpolated time series is longer than the original one. Thus, the forecasting has been conducted based on the augmented time series sequence.

**Strengths:**

1. The idea is novel. The interpolation of the time series in the frequency domain provides some insights for this community.

2. The model efficiency is quite amazing, only 10K parameters have provided the comparable even better performance for the time-series prediction task.

**Weaknesses:**

1. The experiments are not promising. What's the number of the random runs and did you control the random seeds? I suggest to report the mean and std values of the results.

2. As this is the new architecture of the time-series model, more analysis should be conducted such as more prediction tasks, more ablation studies for the key components of the method, etc.

3. The reason behind the effectiveness of the proposed method remains unclear to me.

**Questions:**

N/A.

**Limitations:**

See those in weakness part.

Moreover, the augmented time series is not appropriate for the forecasting tasks. Forecasting is the extension for one given time series, however, simply interpolating for augmenting the original time series is not intuitively correct, because it has modified the original data piece rather than prediction upon the given information (in time domain).
And the experimental results show that the performance is not comparable to the baseline methods such as PatchTST. I suggest the authors to further improve the method and figure out the reason of its effectiveness.

---

> ### Author Rebuttal · Authors · 2023-08-09
>
> - W1: The experiments are not promising. What's the number of the random runs and did you control the random seeds? I suggest to report the mean and std values of the results.
>
> A: *We report the mean and std values across multiple runs in Section B.3 of the Appendix. Tab.4 shows that FITS is very robust to the random seed selection thanks to the small number of parameters. The result shows that over four runs, FITS only shows a 0.001% performance fluctuation.*
>
> - W2: As this is the new architecture of the time-series model, more analysis should be conducted such as more prediction tasks, more ablation studies for the key components of the method, etc.
>
> A: *FITS is a very compact model. Removing any of its parts will make it functional. And FITS only has one hyperparameter for tuning, which is the cutoff frequency. We have done comprehensive ablation studies on the cutoff frequency. Please see the Tab.5 of the manuscript and Section C of the appendix.*
>
> - W3: The reason behind the effectiveness of the proposed method remains unclear to me.
>
> A: *We train FITS to generate an extended time series segment by interpolating the frequency representation of the input time series segment, motivated by the fact that a longer time series provides a higher frequency resolution in its frequency representation. We use a complex-valued linear layer to learn such interpolation. **Please check the Method section of the original manuscript for more information.***
>
> As declared in the Abstract, FITS conducts interpolation on the frequency domain instead of the time domain. Interpolation on the frequency domain, not interpolation on the time domain. Frequency domain interpolation keeps the information in the original data piece and adds frequency for more detail. Please check the Method section of the manuscript for more details.
>
> We also need to emphasize that we **outperform baselines in almost half of the settings** and achieve mostly the second-best in the rest of them. Please check Tables 2 and 3 carefully. Furthermore, our emphasis is on efficiency, not solely on superior performance. Such results align with our major claim that '**achieves comparable or even superior performance to SOTA methods**'.

---

> > ### Comment · Reviewer_pZTr · 2023-08-18
> > **The author has not addressed my concerns.**
> >
> > > We report the mean and std values across multiple runs in Section B.3 of the Appendix.
> >
> > Comparisons between different methods require multi-runs, which will ensure the reproducibility of the experiments and the statistical significance of the comparison. I know that your method is robust to randomness, while I think the common approach is to conduct multiple runs for every compared method to reflect the statistical significance.
> >
> > > Frequency domain interpolation keeps the information in the original data piece and adds frequency for more detail. Please check the Method section of the manuscript for more details.
> >
> > I know you are interpolating in frequency domain. My major point is that interpolation on frequency domain may also influence the input part of the sequence, which is the side effect except forecasting the future horizon. From this perspective, the intrinsic objective is not forecasting, but the disturbance on the original sequence. That's why I do not buy in the motivation and intuition of your method on this task.
> >
> > > our emphasis is on efficiency, not solely on superior performance. Such results align with our major claim that 'achieves comparable or even superior performance to SOTA methods'.
> >
> > From Table 2, the experiments are all conducted on the same dataset with different time spans, and the results are comparable to the baseline(s) in different settings, without obvious trend or characteristics. In Table 3, the proposed method fell back to PatchTST in two datasets with a large margin (in those datasets, difference about 0.01 is very significant). I don't think these results indicated "comparable performance". Yet the comparable one is the DLinear. However, as far as I'm aware, DLinear reported their results only on 1 random seed, which is not promising (see issue #33 in their repository).
> >
> > Before you are claiming about much better efficiency, the performance should be first considered. I think the comparison should be more comprehensive, more promising with more random seeds on all the compared methods.

---

> > > ### Author Response · Authors · 2023-08-19
> > >
> > > We sincerely thank the reviewer for taking the time to respond to our rebuttal. We address these concerns as follows:
> > >
> > > 1. Comparisons between different methods require multi-runs, which will ensure the reproducibility of the experiments and the statistical significance of the comparison. I know that your method is robust to randomness, while I think the common approach is to conduct multiple runs for every compared method to reflect the statistical significance.
> > >
> > >  A: We agree with the reviewer and we shall show the comparison results of multiple runs in the revised version. In the meantime, we would like to point out, *it is not the common practice in the time series forecasting literature*, and we report the results following previous works in this domain.
> > >
> > > 2. My major point is that interpolation on frequency domain may also influence the input part of the sequence, which is the side effect except forecasting the future horizon. From this perspective, the intrinsic objective is not forecasting, but the disturbance on the original sequence. That's why I do not buy in the motivation and intuition of your method on this task.
> > >
> > >  A: Unfortunately, we do not fully understand this comment. Nevertheless, we would like to address it from the following two aspects:
> > >
> > > - For the forecasting task, we care about the forecasting results only, and all techniques (including convolution-, MLP-, and Transformer-based solutions) would try to manipulate the inputs for feature extraction to achieve this objective.
> > >
> > > - More importantly, FITS ****DO PRESERVE**** the information in the input by conducting frequency interpolation. For example, a 16-point segment of a sinusoidal wave with a period of 8. The frequency representation should appear as [0, 0, 1+j, 0, 0, 0, 0, 0, 0]. Suppose we forecast the following 16 steps. After frequency interpolation, the frequency representation would be [0, 0, 0, 0, (1+j)*2, 0, 0, 0, 0, 0, 0, 0, 0, 0, 0, 0, 0]. This is still a sinusoidal wave with a period of 8 and the input part stays intact. For complex waveforms, we can always transform them as the summation of a number of sinusoidal waves, and the above explanation holds true. For detailed results and visualization, please check the ‘Interpretability.ipynb’ in our anonymous code repository link.
> > >
> > > 3. From Table 2, the experiments are all conducted on the same dataset with different time spans, and the results are comparable to the baseline(s) in different settings, without obvious trend or characteristics. In Table 3, the proposed method fell back to PatchTST in two datasets with a large margin (in those datasets, difference about 0.01 is very significant). I don't think these results indicated "comparable performance". Yet the comparable one is the DLinear. However, as far as I'm aware, DLinear reported their results only on 1 random seed, which is not promising (see issue #33 in their repository).
> > >
> > >  A: Due to the space limitation, we break the results into two tables: Table 2 and Table 3. As can be seen in these 2 tables, FITS achieves the ****best performance in 13 out of 28**** settings, and achieves the ****second best in 11 out of 28**** settings. It fails to do so in ****only 4 out of 28**** settings. While PatchTST achieves 15 best and 8 second best. Therefore, we claim that FITS achieves 'comparable' performance. Generally speaking, the channel independence design of PatchTST makes it excel at handling datasets with many variables such as the Electricity (321 channels) and Traffic (862 channels) datasets. In contrast, FITS is suitable to handle datasets with complex periodical patterns such as ETTm2 and weather.
> > >
> > > In our humble opinion, if there exists a model that excels for only one particular type of time series, it has its own value in practice. Therefore, we believe FITS is a promising solution for many practical scenarios.
> > >
> > > 4. Before you are claiming about much better efficiency, the performance should be first considered. I think the comparison should be more comprehensive, more promising with more random seeds on all the compared methods.
> > >
> > >  A: We agree with the reviewer that performance is a critical factor, please refer to our answer to the previous question. At the same time, ****we would like to emphasize that efficiency can be quite critical in practice, especially for edge devices.**** FITS provides very good performance with orders of magnitude smaller compute and memory requirements than existing solutions, making deep learning-based solutions viable for many practical scenarios for the first time, e.g., smart sensors.

---

### Official Review · Reviewer_S88N · 2023-07-06

**Soundness:** 3 good
**Presentation:** 3 good
**Contribution:** 3 good
**Rating:** 6
**Confidence:** 4

**Summary:**

This paper proposes an impressive compact model, named FITS, for time series tasks, including forecasting and anomaly detection. FITS achieves manipulation to time series through interpolation in the frequency domain. The whole framework is quite simple and has remarkably few parameters. FITS achieves competitive performance to SOTA baselines on both forecasting and anomaly detection with about 50 times fewer parameters. With such impressive performance, the proposed model would have a certain impact on the community.

**Strengths:**

1. The idea of implementing time series forecasting and anomaly detection through interpolation in frequency domain is interesting and technically sound.
2. Detailed designs, including LPF and utilization of RevIN, are well motivated and described, making the whole framework reasonable and easy to follow.
3. The proposed model achieves impressive performance on both forecasting and anomaly detection with a remarkably compact size.

**Weaknesses:**

1. Lack of time consumption analysis. Due to rFFT and irFFT in the model, time efficiency is the main concern.
2. The coverage of related works is barely satisfactory. Adding some preliminary about manipulation in frequency domain to the manuscript would be helpful to understand the model.
3. Typos. For example, duplicated citations in line 383 and line 385.

**Questions:**

1. It is appreciated to further list time consumption of the proposed model under different settings and compare time efficiency with baselines.
2. For forecasting task, what if the model is supervised in frequency domain?
3. Did authors try other interpolation methods, e.g., convolution?

**Limitations:**

Not applicable.

---

> ### Author Rebuttal · Authors · 2023-08-04
>
> Thanks for your positive evaluation of our work and the insightful questions.
>
> - Q1, W1: It is appreciated to further list time consumption of the proposed model under different settings and compare time efficiency with baselines. Due to rFFT and irFFT in the model, time efficiency is the main concern.
>
> A: *We conduct experiment to measure the time consumption and find FITS can provide **sub-millisecond-level inference time**. **Please refer to the 'About Efficiency' section of the common response for detailed analyze.***
>
> *We report the comparison with baselines as below. We conduct experiment on a NVIDIA Titan Xp and follow DLinear's setting, Electricity dataset, 720 input and 96 output. **The result labeled with CPU is run on a single core CPU to simulate the scenario on resources limited edge device**.*
>
> |Model|Inference Time|
> |:---:|:---:|
> |FITS|0.6ms|
> |FITS(CPU)|2.55ms|
> |DLinear|0.4ms|
> |DLinear(CPU)|3ms|
> |Informer|49.3ms|
> |Autoformer|164.1ms|
> |Pyraformer|3.4ms|
> |FEDformer|40.5ms|
>
>
>
> - Q2: For forecasting task, what if the model is supervised in frequency domain?
>
> A: *This is an insightful question and it is our **initial idea**. However, the frequency domain is a complex domain, and there is no effective differentiable loss function on the complex domain. Furthermore, supervising the real and imaginary parts of complex numbers is not feasible, breaking the phase information in the complex value.*
>
> - Q3: Did authors try other interpolation methods, e.g., convolution?
>
> A: *Thanks for the suggestion and we will try more in our future work. In our current opinion, convolution may not be a feasible interpolation method. According to the property of Fourier Transform, convolution on the frequency domain equals the multiplication on the time domain which may not properly manipulate the input to get the ideal result.*
>
> - W2: The coverage of related works is barely satisfactory. Adding some preliminary about manipulation in frequency domain to the manuscript would be helpful to understand the model.
>
> A: We put the *‘Preliminary: FFT and Complex Frequency Domain’ part in the Method Section to introduce the basis of the complex frequency domain and the calculation and its corresponding physical meaning.* We would expand this part and add a separate preliminary section in the revised version.
>
> - W3: About typos.
>
> A: *We will carefully proofread our paper and fix it in the revised version.*

---

> > ### Comment · Reviewer_S88N · 2023-08-13
> >
> > The authors have addressed my concerns well and I will keep the score as it is.

---

> > > ### Author Response · Authors · 2023-08-18
> > > **Thanks for your assessment!**
> > >
> > > Thanks for your assessment of our work! Please feel free to ask if you have further question!

---

### Official Review · Reviewer_P7AV · 2023-07-08

**Soundness:** 3 good
**Presentation:** 3 good
**Contribution:** 3 good
**Rating:** 7
**Confidence:** 4

**Summary:**

The authors present a transformer model for forecasting and anomaly detection. The model performs both forecasting and reconstruction in the frequency domain with a fraction of parameters compared to state-of-the-art transformers. The model demonstrates impressive performance on anomaly detection and forecasting tasks.

**Strengths:**

1. This is an interesting work and novel to the best of my knowledge.
2. I strongly believe that the lightweight nature of the model is an asset.
3. The model demonstrates near state-of-the-art performance.
4. The paper is well-written.

**Weaknesses:**

Major: Most of these are not deal breakers, but the following would make the evaluation of the model stronger:
1. Baselines for forecasting : The authors compare with only transformer based models. I would encourage the authors to compare with advanced non-transformer based models such as N-HiTS [1] and N-BEATS [2].
2. Handling of multi-variate time-series data (see Q1)
3. Baselines for anomaly detection: I would again encourage the authors to compare with state-of-the-art time-series anomaly detection e.g., DGHL [3].
4. Evaluation for anomaly detection detection: I would encourage the authors to use standard evaluation metrics for anomaly detection like adjusted best F1 [3, 4], Average Precision [4, 5], and Volume Under Surface (VUS) [6]. The datasets used also have known flaws [7].

Minor:
1. "To avoid information leakage, We choose" --> "To avoid information leakage, we choose"

References:

1. Challu, Cristian, et al. "NHITS: Neural Hierarchical Interpolation for Time Series Forecasting." Proceedings of the AAAI Conference on Artifi
2. Oreshkin, Boris N., et al. "N-BEATS: Neural basis expansion analysis for interpretable time series forecasting." arXiv preprint arXiv:1905.10437 (2019).
3. Challu, Cristian I., et al. "Deep generative model with hierarchical latent factors for time series anomaly detection." International Conference on Artificial Intelligence and Statistics. PMLR, 2022.
4. Goswami, Mononito, et al. "Unsupervised model selection for time-series anomaly detection." arXiv preprint arXiv:2210.01078 (2022).
5. Schmidl, Sebastian, Phillip Wenig, and Thorsten Papenbrock. "Anomaly detection in time series: a comprehensive evaluation." Proceedings of the VLDB Endowment 15.9 (2022): 1779-1797.
6. Paparrizos, John, et al. "Volume under the surface: a new accuracy evaluation measure for time-series anomaly detection." Proceedings of the VLDB Endowment 15.11 (2022): 2774-2787.
7. Wu, Renjie, and Eamonn Keogh. "Current time series anomaly detection benchmarks are flawed and are creating the illusion of progress." IEEE Transactions on Knowledge and Data Engineering (2021).

**Questions:**

1. How does the model handle multi-variate time-series data? Does the model capture cross-channel information?
2. What are MACs (Multiply-Accumulate Operations)? Neither the paper nor the supplementary material talks about it.
3. I would like to know exactly how the reconstruction works, I am not sure if I understand it with the given information.

**Limitations:**

The authors discuss some limitations of your work.

---

> ### Author Rebuttal · Authors · 2023-08-04
>
>
> - W1: Baselines for forecasting : The authors compare with only transformer-based models. I would encourage the authors to compare with advanced non-transformer-based models such as N-HiTS and N-BEATS.
>
> A: *We show the comparison with mentioned N-HiTS and N-BEATS on MSE in the following table. FITS outperforms these two models in most cases while maintaining a compact model size. We will consider adding the following results to our main result. The results for N-HiTS and N-BEATS are retrieved from the paper of N-HiTS[1].*
>
> |   Dataset   | Horizon |  FITS  | N-BEATS | N-HiTS |
> | :---------: | :-----: | :----: | :-----: | :----: |
> | Electricity |   96    | **0.138** | 0.145  | 0.147 |
> |             |   192   | **0.152** | 0.180  | 0.167 |
> |             |   336   | **0.166** | 0.200  | 0.186 |
> |             |   720   | **0.205** | 0.266  | 0.243 |
> |   Traffic   |   96    | 0.401 | **0.398**  | 0.402 |
> |             |   192   | **0.407** | 0.409  | 0.420 |
> |             |   336   | **0.420** | 0.449  | 0.448 |
> |             |   720   | **0.456** | 0.589  | 0.539 |
> |   Weather   |   96    | **0.145** | 0.167  | 0.158 |
> |             |   192   | **0.188** | 0.229  | 0.211 |
> |             |   336   | **0.236** | 0.287  | 0.274 |
> |             |   720   | **0.308** | 0.368  | 0.351 |
> |    ETTm2    |   96    | **0.164** | 0.184  | 0.176 |
> |             |   192   | **0.217** | 0.273  | 0.245 |
> |             |   336   | **0.269** | 0.309  | 0.295 |
> |             |   720   | **0.347** | 0.411  | 0.401 |
>
>
> *Note that we also compare FITS with the linear-based model (DLinear[2]) and convolution-based model TimesNet. Furthermore, we also show the comparison with the aforementioned N-HiTS and N-BEATS on the M4 dataset. These results are shown in  Table 3 of the appendix.*
>
>
> - W2&Q1: How does the model handle multi-variate time-series data? Does the model capture cross-channel information?
>
> A: *FITS employs weight sharing across channels, a commonly adopted approach that balances performance and efficiency. Moreover, individual FITS models per channel were experimented with, yet no substantial performance improvement was observed.*
>
> *FITS does not have a specific cross-channel information-capturing mechanism. But it shows competitive performance. We believe that introducing such a mechanism could enhance FITS' performance, although this may introduce additional parameters.*
>
> *Please refer to the **'About Multi-variate' section of common response** for detailed analysis.*
>
> - Q2: What are MACs (Multiply-Accumulate Operations)? Neither the paper nor the supplementary material talks about it.
>
> A: *Sorry for using the abbreviation without explanation. We will add the explanation in the revised version. MACs (Multiply-Accumulate Operations) is a commonly used metric that counts the total number of multiplication and addition operations in a neural network. We follow the DLinear[2] to use MACs as our metric to measure computational efficiency.*
>
> - W3&W4: Baselines for anomaly detection and Evaluation.
>
> A: *We show the comparison with the mentioned DGHL in the common response and will add it to our main result table in the revised version. FITS outperforms the DGHL on SWaT dataset by a large margin. On the SMAP and MSL datasets, FITS shows suboptimal results. As analyzed in the original manuscript, FITS is not suitable for handling binary data in these two datasets.*
>
> *We design our anomaly detection experiment following the methodology of the Anomaly Transformer. For fair and convenient comparison, we adopt the same metrics and benchmarks. We will add the mentioned metrics and benchmarks to the revised version of the paper.*
>
> - Q3: I would like to know exactly how the reconstruction works, I am not sure if I understand it with the given information.
>
> A: Please refer to Fig. 1 in the **Supplementary Material of this paper** for an illustration of the **reconstruction pipeline**. In this process, the model input $x$ is derived from a segment of the time series $y$ using an **equidistant sampling** technique with a specified downsample rate $\eta$. Subsequently, FITS performs frequency interpolation, generating an upsampled output* *$\hat{x}_{up-sampled}$* *with the same length as $y$. The reconstruction loss is computed by comparing the original $y$ and the upsampled* $ \hat{x}_{up-sampled}$. *Please note that, due to space constraints, the depicted downsample/upsample rate $\eta$ in the figure is shown as 1.5, which is not a practical value. In our actual experiments, we employ a $\eta$ value of 4.*
>
> [1] NHITS: Neural Hierarchical Interpolation for Time Series Forecasting. AAAI 2023.
>
> [2] Are transformers effective for time series forecasting? AAAI 2023.

---

> > ### Comment · Reviewer_P7AV · 2023-08-13
> > **Thanks for the response!**
> >
> > I would like to thank the authors for their response! I would like to bump up the score a bit to reflect my current assessment of the paper.

---

> > > ### Author Response · Authors · 2023-08-18
> > > **Thanks for your assessment of our work!**
> > >
> > > Thanks for your assessment of our work! Please feel free to ask if you have further question!

---

### Author Rebuttal · Authors · 2023-08-04

We deeply appreciate the reviewers for their insightful feedback and recognition of our work's strengths:

1. The concept of conducting interpolation after projecting time series into the frequency domain is both **innovative and technically sound** [Reviewers PA7V, S88N, dtUP, pZTr].
2. The comprehensive explanation of our detailed designs has been acknowledged for making the entire framework **coherent and easily comprehensible** [Reviewers S88N, o3Wj].
3. FITS' **exceptional efficiency** has been highlighted, as it achieves remarkable performance in forecasting and anomaly detection tasks while maintaining a remarkably compact size [Reviewers PA7V, S88N, pZTr, dtUP, o3Wj].
4. Our experimental results **effectively validate our assertions**, showcasing competitive performance and a significant reduction in model complexity compared to existing SOTA methods [Reviewers o3Wj, dtUP, PA7V].
5. FITS shows **great practical value** because of its compact model size. This contribution makes it **great application for actual real-world scenarios**, applying on edge devices. [Reviewers PA7V, o3Wj]

Following are the responses to **common questions** raised by reviewers. We will add them in our final version.

---

### G1: Handling Multi-channel data. [Reviewer o3Wj, P7AV]

1. Can FITS handle multivariate series? [Reviewer **o3Wj**]

***Yes, FITS is capable of handling multivariate time series data.** Our experiments were conducted on **eight** multivariate datasets, and the dimensions of the datasets are shown in Table 1 of the original paper.*

2. How does the model handle multi-variate time-series data? [Reviewer **P7AV**]

*FITS employs weight sharing across multi-variates/dimensions/channels, a commonly adopted approach that balances performance and efficiency. Moreover, individual FITS models per channel were experimented with, yet no substantial performance improvement was observed.*

*Additionally, we note that channels often share a common base frequency when originating from the same physical system. For instance, signals from circuits commonly have a base frequency of 50/60Hz, while traffic flow across a city follows a daily base frequency. This observation further supports the suitability of the weight-sharing strategy for FITS.*

3. Does the model capture cross-channel information? [Reviewer **P7AV**]

*FITS does not have a specific cross-channel information-capturing mechanism. But it shows competitive performance. We believe that introducing such a mechanism could further enhance FITS' performance, but it would lead to heavier models compared to its current form.*

4. How to choose cutoff frequency in case of multivariate series, when the harmonics are likely to be different over variables? [Reviewer **o3Wj**]

*The harmonic cutoff is determined by observing the amplitude spectrum and selecting the largest harmonic across the dataset. Our experiments show that performance gains from raising the cutoff frequency become minor beyond the second/third order harmonic.*

*Regarding different base frequencies, our observations suggest that channels with distinct base frequencies generally have minor correlations. Nevertheless, designers can specify the cutoff frequency for each channel to adapt the data's specific characteristics.*

---

### G2: About the efficiency:

To show extraordinary efficiency, we compare FITS with DLinear. As a current SOTA baseline with a single-layer Linear network, DLinear is indeed quite efficient, but it might still be too large to run on MCU-powered edge devices.

1. About the time consumption and the impact of rFFT and irFFT: [Reviewer **S88N**]

*Our experiments show that FITS can provide **sub-millisecond-level inference time**. Following DLinear, we evaluate the inference time of FITS on the Electricity dataset with input length 720 and output length 96. On one NVIDIA Titan Xp, the inference time of FITS is **0.6ms** and DLinear is 0.4ms. Moreover, the inference time on a **single core of CPU** of FITS is **2.55ms**, which is better than DLinear's 3ms. The result is averaged over 5164 runs (over the entire test set).*

*In our experiment, we find rFFT and irFFT together only introduce 0.7ms time overhead (on CPU). In practice, we can offload FFT to dedicated chips such as FPGA or DSP, which can conduct such algorithms faster. Under this scenario, FITS finally achieves 1.7ms inference time on a single-core CPU. Such time consumption is usually negligible compared with communication latency.*

2. About the training efficiency of FITS: [Reviewer **dtUP**]

*FITS exhibits **remarkable training efficiency**, requiring only around 5 minutes for ETT datasets and 30 minutes for the traffic dataset (with 862 channels) on a single NVIDIA Titan Xp. Intriguingly, for both FITS and DLinear, the training bottleneck is CPU loss computation rather than the model itself. FITS' compact size also ensures a minimal memory footprint.*

---

### G3: About Anomaly Detection: [Reviewer **o3Wj, S88N, dtUP, P7AV**]

*We compare with the baselines mentioned by reviewers as below (F1-score). While the results are quite encouraging, we want to emphasize that we conduct AD to demonstrate the use case of FITS **instead of claiming its superior performance.***

|Dataset|FITS|DGHL|OCSVM|IForest|
|:-:|:-:|:-:|:-:|:-:|
|SMD|**99.95**|N/A|56.19|53.64|
|PSM|**93.96**|N/A|70.67|83.48|
|SWaT|**98.9**|87.47|47.23|47.02|
|SMAP|70.74|**96.38**|56.34|55.53|
|MSL|78.12|**94.08**|70.82|66.45|

*Designed as a lightweight algorithm for edge devices, FITS offers significant benefits with minimal memory and computational demands. Impressively, it achieves **sub-millisecond** inference times, negligible in contrast to large models or communication. This renders FITS ideal for **swiftly detecting critical errors**. Integrated as a coarse-grained filter with a specialized AD algorithm for finer-grained detection, the system ensures robustness and responsiveness to a range of anomalies.*

---

### Decision · Program_Chairs · 2023-09-21

**Decision:**

Reject

**Comment:**

The paper proposes a method on time-series forecasting and anomaly detection. Although the paper has some interesting ideas and results, the reviewer finds that the paper has some practical issue that the experiment has not been conducted with multiple runs, which makes it less promising. The reviewer also mentioned that as for the experiments, the results are not persuasive that this model is effective enough on the current benchmarks. Moreover, they think that the authors has over-claimed the “comparable performance” of the proposed method.